# Mobile measurement of methane emissions from natural gas developments in Northeastern British Columbia, Canada

Emmaline Atherton[1], David Risk[1], Chelsea Fougere[1], Martin Lavoie[1], Alex Marshall[1], John Werring[2], James P. Williams[1], and Christina Minions[1]

[1]Department of Earth Sciences, St. Francis Xavier University, Antigonish, Nova Scotia, Canada B2G 2W5
[2]David Suzuki Foundation, Vancouver, British Columbia, Canada V6K 4R8

*Correspondence to:* Emmaline Atherton (eatherto@stfx.ca)

**Abstract.** North American leaders recently committed to reducing methane emissions from the oil and gas sector, but information on current emissions from upstream oil and gas developments in Canada are lacking. This study examined the occurrence of methane plumes in an area of unconventional natural gas development in northwestern Canada. In August to September, 2015 we completed almost 8 000 km of vehicle-based survey campaigns on public roads dissecting oil and gas infrastructure, such as well pads and processing facilities. We surveyed six routes 3-6 times each, which brought us past over 1 600 unique well pads and facilities managed by more than 50 different operators. To attribute on-road plumes to oil and gas related sources we used gas signatures of residual excess concentrations (anomalies above background) less than 500 m downwind from potential oil and gas emission sources. All results represent emissions greater than our minimum detection limit of 0.59 g/s at our average detection distance (319 m). Unlike many other oil and gas developments in the US for which methane measurements have been reported recently, the methane concentrations we measured were close to normal atmospheric levels, except inside natural gas plumes. Roughly 47% of Active wells emitted methane-rich plumes above our minimum detection limit. Multiple sites that pre-date the recent unconventional natural gas development were found to be emitting, and we observed that the majority of these older wells were associated with emissions on all survey repeats. We also observed emissions from gas processing facilities that were highly repeatable. Emission patterns in this area were best explained by infrastructure age and type. Extrapolating our results across all oil and gas infrastructure in the Montney area, we estimate that the emission sources we located (emitting at a rate > 0.59 g/s) contribute more than 111 800 tonnes of methane annually to the atmosphere. This value exceeds reported bottom-up estimates of 78 000 tonnes methane for all oil and gas sector sources in British Columbia. Current bottom-up methods for estimating methane emissions do not normally calculate the fraction of emitting oil and gas infrastructure with thorough on-ground measurements. However, this study demonstrates that mobile surveys could provide a more accurate representation of the number of emission sources in an oil and gas development. This study presents the first mobile collection of methane emissions from oil and gas infrastructure in British Columbia, and these results can be used to inform policy development in an era of methane emission reduction efforts.

# 1 Introduction

As global energy needs continue to rise, oil and gas operators are increasingly recovering natural gas from less-permeable reservoirs, such as tight sandstone and shale, despite environmental concerns surrounding extraction methods. Unconventional techniques, such as horizontal drilling and multi-stage hydraulic fracturing, can be used to stimulate production of natural gas directly from the source-rock in a petroleum system, ultimately increasing the total quantity of marketable natural gas. Presently, Canada is the fifth-largest producer of natural gas worldwide, with enough unrecovered natural gas to sustain 2013 national consumption levels for 300 years (NEB, 2016). More than 68% of Canada's remaining 1087 trillion cubic feet of marketable natural gas reserves is in unconventional reservoirs (NEB, 2016). By 2035, Canadian natural gas production is predicted to increase 25% above 2013 levels, and this projected growth is largely attributed to unconventional methods of extraction such as horizontal drilling and multi-stage hydraulic fracturing.

Compared to coal, natural gas is often considered to be a preferable fossil fuel because it emits 50-60% less carbon dioxide ($CO_2$) during combustion (NETL, 2010). As such, natural gas has been deemed a transition fuel on the path to renewable energy because it allows for continued fossil fuel exploitation while seemingly emitting a smaller amount of greenhouse gases. However, the primary component of natural gas is methane ($CH_4$), a very potent greenhouse gas (GHG), so leaks of natural gas directly to the atmosphere contribute to climate change. The radiative forcing of $CH_4$ is greater than 30 times that of $CO_2$ over a 100-year timespan (IPCC, 2014). A recent study suggests that if more than 3.2% of total natural gas production is emitted into the atmosphere during upstream operations, the environmental benefit of combusting natural gas, instead of coal or oil, is negated (Alvarez et al., 2012). Therefore, to comprehensively analyze the GHG footprint of different fuel types, it is necessary to consider industrial emissions during upstream operations; these include both vented (intended) and fugitive (unintended) emissions from wells, facilities, and pipelines, during extraction, production, and processing.

"Well-to-wheel" life-cycle assessments (LCA) are a method of comparing the environmental impact of fossil fuels in relation to their carbon emissions. This type of LCA sums all estimated carbon outputs, including emissions during upstream operations, transportation, and combustion. Several recent LCAs suggest that the carbon footprints of unconventional natural gas developments exceed those of conventional natural gas developments (primarily due to emissions during well completions), but that coal developments have the worst overall emissions impact (Hultman et al., 2011; Jiang et al., 2011; Skone, 2011; Stephenson et al., 2011). Contrastingly, another study suggests that conventional natural gas has a slightly higher carbon footprint than unconventional natural gas because of emissions during the liquid unloading process, but that coal remains the fossil fuel with the highest life-cycle carbon emissions (Burnham et al., 2012). A controversial study by Howarth et al. (2011) concluded that a large amount of atmospheric emissions associated with upstream shale gas operations render its environmental impact more severe than coal. This study has been widely disputed for overestimating $CH_4$ emissions during upstream shale gas processes by not acknowledging that gases emitted during well completions are often flared or controlled by performing reduced emission completions (CNGI et al., 2012). The variability of results from these recent "well-to-wheel" LCAs demonstrates that total upstream emission volumes are difficult to quantify using estimated emission frequencies for infrastructure. It is important to know what percentage of infrastructure is actually emitting, and active detection and measuring techniques are

required to gain this understanding. Furthermore, it is important to note that emission frequencies may vary between oil and gas developments because of operator best practice, or due to the properties of the geological formation that the hydrocarbons are being extracted from. (In this paper, "development" refers to areas of hydrocarbon extraction, and "infrastructure" refers to oil and gas related infrastructure such as well pads and processing facilities).

The common infrastructural sources of fugitive emissions are poorly understood, particularly in unconventional natural gas developments where these extraction practices are newly implemented. Detection of atmospheric fugitive emissions from upstream sources has previously been attempted with top-down methods and specific ground-based techniques. Top-down measurements include airborne (Karion et al., 2013; Caulton et al., 2014), and remote sensing (Govindan et al., 2011; Schneising et al., 2014) measurements. These methods often cover large areas in low resolution proving difficult to identify exact sources of
emissions. Ground-based techniques, including infrared camera leak inspections (Mitchell et al., 2015), well injection tracers (Mayer et al., 2013), and soil gas sampling (Beaubien et al., 2011; Romanak et al., 2012), are often too labour intensive to be convenient for use in large oil and gas developments.

Although a recent study assumes that around 63% of infrastructure is emitting in the Barnett Shale (Rella et al., 2015), the majority of inventory studies do not report the occurrence of emitting and non-emitting infrastructure. Ultimately, $CH_4$
management will entail a coordinated targeting of emission sources and reduction of overall emission frequency. So, studies that build geospatially distributed information on emission frequencies in large populations of infrastructure is a logical next step, because it is the best means of identifying trends across vast developments, behavioural patterns of operators, and the impact of infrastructure age on emission frequency and severity. Mobile screening methods similar to EPA OTM33A (Brantley et al., 2014), even that simply detect emission frequencies, are extremely valuable because emission factors are already available
and can be applied uniquely to known emitters so that volumes can be estimated to a reliable degree.

In this study we used a multi-gas ($CO_2$, $CH_4$) mobile surveying method that uses ratio-based gas concentration techniques and wind data to detect and attribute on-road $CH_4$-rich plumes to the infrastructural sources of natural gas developments in northeastern British Columbia, Canada. Our primary interest in this study was to determine the frequency of emissions, and the relationship between emissions and specific classes of infrastructure. We applied this method in an area that is commonly
referred to as the Montney, in reference to the extensive, petroleum-rich, geologic formation covering 130 000 km$^2$ aerially between British Columbia and Alberta (BC Oil and Gas Commission, 2013). It was first recognized as an unconventional petroleum reservoir in 2007, and attempts at accessing its resources were accomplished with horizontal drilling and multistage hydraulic fracturing. These unconventional methods yielded 4-5 times more natural gas from the Montney formation than conventional techniques that were attempted prior to 2005. Since then, production of BC unconventional natural gas has
increased significantly, with the Montney play being the largest contributor (BC Oil and Gas Commission, 2012).

While the Montney is a challenging first target for mobile emissions surveying because of its spatial extent and lack of accessibility (many poor condition roads), it is a sensible first choice given that its emissions have not been measured independently of industry and government, and because the production mode is largely unconventional - and therefore subject to a higher degree of scrutiny. The less permeable, natural-gas hosting portion of the Montney formation is located in BC, a province that

has generally been very progressive on many issues of environmental stewardship, so there is a broad interest in emissions quantification and environmental performance.

## 2 Methods

### 2.1 Field Measurements

Between August 14, 2015 and September 05, 2015, we collected atmospheric gas concentration data along six pre-planned routes in the Montney formation of northeastern BC (Fig. 1). We designed the routes to come as close as possible to a high number of unconventional natural gas wells and their associated processing facilities, while also incorporating a variety of operators and infrastructure age profiles. These were on-road campaigns only, and did not approach well pad infrastructure or facilities.

In total we surveyed 7 965 km of public roads, with an average route length of 248 km (Table 1). We collected gas concentrations at 1 Hz frequency while surveying. The Regional Route and Routes 2, 3, 4 (Fig. 1) dissected natural gas developments containing unconventional natural gas wells. Route 1 targeted an older development in the same area that mainly produces oil; this route was intended for preliminary comparison between conventional oil and unconventional natural gas developments. The Control Route was located outside the perimeter of concentrated natural gas infrastructure, and was intended to act as

a control. We surveyed four of the routes (Routes 1, 2, 3, 4) six times throughout the field campaign, and the two remaining routes (Regional Route and Control Route) three times each. We repeated surveys on multiple days to account for varying wind directions. Repetitions of each survey route included both morning and afternoon drives to incorporate varying atmospheric conditions. We also used the repeated survey data to obtain statistics on emission persistence throughout our 23-day survey campaign.

The mobile surveying platform we used to collect these data consisted of an LGR Ultraportable Greenhouse Gas Analyzer (Los Gatos Research Inc., San Jose, CA, USA) Off-Axis Integrated Cavity Output Spectrometer ($1\sigma$ instrumental errors of < 2 ppb at 1 sec), to measure raw atmospheric concentrations of $CO_2$, $CH_4$, and $H_2O$. A high volume (7 lpm) air pump brought air to the analyzer from the front of the vehicle through 6 mm ID tubing. During post-processing we applied corrections for lag times between the intake filter and the gas analyzers. An NM 150 weather station (New Mountain Innovations, Old Lyme,

CT, USA) was located 1.5 metres above the vehicle to collect wind and weather data (with instrumental errors of $\pm$ 1.5° for wind direction and $\pm$ 4% for wind speed (measured in km/h)). Gas species concentrations and wind velocity measurements were collected every second while driving. Wind velocity measurements were corrected for both the direction and speed of the vehicle, and we geo-located all data-points using a handheld Garmin GPS. We stored all observations in a database, with processing, statistics, and plots completed in R (R Core Team, 2016).

## 2.2 Identification of Natural Gas Emissions

Both $CO_2$ and $CH_4$ exist, and vary, naturally in the atmosphere. We had to account for this variance in order to identify anomalous measurements that were potentially sourced from natural gas developments. Variation of $CO_2$ within the survey area was likely primarily a function of oilfield processes (emissions, engines, flares) because there was little industrial activity on the survey routes that was not related to oil and gas development.

To accommodate the fluctuating background concentrations of $CO_2$ and $CH_4$, the traditional approach would either be for the user to set a concentration threshold above which a reading would be considered an anomaly, or for a dataset minimum value to be used as the background (as in Hurry et al. (2016)). The survey routes in our study were multiple hours long each and were often routed through various land use types. For this reason, we did not use the traditional methods of calculating background atmospheric gas concentrations. Instead, we used a simple iterative deconvolution method in which we reset the ambient "background" concentration of each gas at a specified time interval, called the Running Minimum Reset Interval (RMRI), and where we iteratively scaled the RMRI until we had maximized the number of (consecutive multi-point) above-background ("*excess*") ratio emission anomalies. In other words, an optimal RMRI was determined for each survey by iteratively applying a suite of RMRI values (60s to 1800s, at an interval of 60s) to our datasets, subtracting the background, and evaluating the number of multipoint $eCO_2$:$eCH_4$ < 150 excursions. As RMRIs shortened, a higher number of small emission anomalies were exposed, by about 2-3 times relative to the dataset minimum approach used by Hurry et al. (2016). However, when the iteration approached very small RMRIs (< 180 s), it consistently caused the total number of anomalies to increase (often by a factor of 10), in particular for anomalies of extremely small concentration. This was expected because when we reset background concentrations too quickly, it overlaps in the temporal domain with instrument and other random noise, causing every departure to seem anomalous relative to the recently reset background. Our algorithms chose the optimal RMRI to be the point at which anomalies were maximized, but also where we avoided the rapid noise-associated increase that we saw with extremely short RMRIs (Fig. 2). We applied this method separately to each of the 30 surveys for both $CO_2$ and $CH_4$ concentrations. RMRIs of about 300 s were normally most favourable for the resolution of $eCO_2$:$eCH_4$ < 150 excursions, but for some surveys in more consistent terrain (or weather) longer RMRIs proved better. This means that for most surveys, our algorithms reset the background concentration for each gas every ~300 s, to the lowest recorded concentration value during the preceding 300 s. While this background subtraction technique improves the resolution of localized plumes, it should be clear that it impedes the resolution of larger regional anomaly features, or mega-plumes, because they may in fact form an artificially elevated background that persists across the 300 s scale. We differentiated occurrences of combustion emissions from other emission sources by filtering out all values where $eCO_2$:$eCH_4$ > 1000. Combustion-related emission sources include vehicle tailpipe emissions and industry (ex. power generation).

We identified $CH_4$ plumes from oil and gas infrastructure in areas where there were multiple successive datapoints with depressed $eCO_2$:$eCH_4$ values. The $CO_2$:$CH_4$ ratio of ambient air is roughly 215, and $CH_4$-rich plumes from natural gas sources are substantially more depressed at the point of origin (the Montney does contain low amounts of $CO_2$ in variable, but generally super-ambient, concentrations). We used ratios of these gases in detection instead of raw $CH_4$ concentrations, because

ratios are more conservative than concentrations in valleys and other areas where pooling of gases is common, and fewer false positives are likely. Since fugitive and vented gas sources might be highly diluted in air, their presence will not significantly affect the normal bulk ratio. In this case, the $e$CO$_2$:$e$CH$_4$ ratio will record the anomalies with a higher degree of fidelity. This eCO$_2$:eCH$_4$ approach has proven to be a useful fingerprinting tool in oil and gas environments because a single ratio value can help elucidate the presence of multiple emission source types. In this study, we follow a procedure similar to Hurry et al. (2016), and a detailed explanation of the method is described there. For our study, we assumed that $e$CO$_2$:$e$CH$_4$ ratios < 150 were representative of significant departures from the normal natural ratio, and potentially indicative of an exogenous CH$_4$ source locally. In order for a natural gas related plume to be identified, we had to detect > 3 successive datapoints with $e$CO$_2$:$e$CH$_4$ ratios < 150.

## 2.3   Emission Source Attribution

We used publicly available files from the BC Oil and Gas Commission (BC OGC) (acquired July, 2015) of all oil and gas infrastructure in the province to attribute the plumes to potential emission sources based on wind direction and distance. We modified these files to exclude temporary or virtual facilities, such as those in place only during well drilling, or artificial facility entries used to record regulatory information. Otherwise, all in-place oil and gas wells and facilities were considered possible emission sources. The infrastructure database included the well and facility locations, as well as various attribute data such as infrastructure types, statuses, and spud dates (drilling dates). In the field, we attempted to verify the locations in the infrastructure database when possible. The locations of the majority of well pads and processing facilities appeared to be accurate, however the statuses in the database may not have been up to date. For example, well pads recorded as "Abandoned" in the database occasionally still had infrastructure present. Although we could not verify the locations of all infrastructural sources from public roads, we assumed based on our experience in the field that infrastructure locations were correct but that there may be discrepancies in the attribute information. When we detected $e$CO$_2$:$e$CH$_4$ < 150 excursions on-road, and infrastructure was present upwind within the target radius of 500 m, our attribution method flagged that infrastructure as a probable emission source.

We did not use a unique thermogenic tracer to discriminate biogenic CH$_4$ sources, such as cattle that may have been present on the well sites at the time of surveying. However, repeated surveying of each route increased our confidence that we were tagging stationary natural gas infrastructural sources. Persistence is also an important metric, not only for detection, but because many of these fugitive and vented emissions are episodic in nature. Though the infrastructure is stationary, the emissions are not necessarily continuous, and gas migrations, surface casing vent flows, leaks, and tank vents, are all known to have a temporal component. Additionally, maintenance activities may have been occurring onsite at the time of survey, which would generate a non-persistent emission pattern and occasionally we were proximal to drilling or fracturing operations. For this reason, this study does not analyze episodic emission sources, so all infrastructure that we identified as "emitting" should be thought of as continual, persistent, emission sources.

## 3  Results and Discussion

We collected atmospheric gas concentration data along 30 surveys of six different routes. The routes ranged in length from 200 - 550 km (Table 1), and, at the time of surveying, more than 50 different operators managed the oil and gas infrastructure located on these routes. Compared to some oil developments in western Canada, natural gas developments in northeastern British Columbia are spread out, and therefore required a considerable amount of driving to survey thoroughly. It was not possible to secure a Control Route that was totally free of oil and gas infrastructure, but our Control route did have a density of infrastructure that was much lower than that of other routes, with intervals that were relatively unpopulated.

### 3.1  Measured Gas Signatures

Methane was the gas of primary interest for this study, and bulk $CH_4$ values were in general not appreciably different from background air. Mean $CH_4$ for the study was 1.897 ppm with $\sigma$=0.084 ppm (n=444515). Max and min were 8.148, and 1.819, respectively. Since the background was very stable, anomalies that we detected near oil and gas infrastructure were both obvious, and short-lived. These bulk concentrations contrast with those measured for other developments. For example a study in the Barnett Shale measured a mean $CH_4$ concentration of 11.99 ppm, with a median of 2.7 ppm, in residential fringes surrounding shale gas development (Rich et al., 2014). The Barnett Shale has about three times as much infrastructure in half the area, but the mean departures in the Barnett exceed the maximum departure in this study. In the Montney, ambient $CH_4$ concentrations were seldom measurably different than global norms (about 1.850 ppm, but regionally dependent). As a result of the stable background, combined with the deconvolution approach, we were able to use the mobile survey approach to detect the presence of emissions hundreds of metres away from infrastructure. On average, we were sampling infrastructure from a distance of 319 m (Figure 3), and we detected natural gas emissions from a mean distance of 314 m (between the point of data collection and the probable emission source).

Figure 4 shows the aggregate (all survey repetitions) kernel density plots of $e$CO$_2$:$e$CH$_4$ for the survey routes (ratios of $CO_2$ to $CH_4$ above ambient). In each density plot, there is a peak near the $e$CO$_2$:$e$CH$_4$ value 220, which is representative of the ratio between ambient $CO_2$ and $CH_4$. Though most of the ambient concentrations should be filtered out in background subtraction, some of the background signature remains in our datasets during the initial increase and decrease in $CH_4$-enriched peaks. The kernel density plots in Figure 1 show that, in all of the survey routes except the Control, we see a population of $CH_4$-enriched anomalies (less than the natural ratio of 220), that are the result of localized plumes from natural gas development. The Control Route lacked an obvious population of enriched $CH_4$ values, which was expected because the density of infrastructure was comparatively low.

We did not see any $CH_4$-rich plumes that would be characteristic of a super-emitter. This is evident by the fact that the maximum raw $CH_4$ value we recorded was low (8.148 ppm). These low emission magnitudes are inline with results from GreenPath Energy  (2017), which used FLIR cameras to assess emission sources in the Alberta portion of the Montney formation.

## 3.2 Emission Sources and Trends

Once we classified plumes based on their geochemical signatures, we attributed them to nearby oil and gas infrastructure. An example of this binary result is presented visually in Figure 5, where infrastructure is shown in red when tagged as emitting, or in green when on-road plumes were absent. However, we rarely dealt with the maps directly because our aim was to investigate industry-wide patterns, and drivers, across types and age classes of infrastructure and operators. For further analysis, these binary data were folded into datasets along with infrastructural characteristics extracted from the geospatial databases. While surveys of the Control Route allow us to be very confident about the existence of plumes, we are less confident about the precise origin of the plume. In areas of low infrastructural density, geospatial attribution confidence is maximized. But in areas of high infrastructure density, it is possible that emissions from a suspected source are actually being emitted from a co-located battery, gathering pipeline, or other. A Forward-Looking Infrared (FLIR) camera would be required to trace each plume precisely to the source gasket, vent, or soil area, and that work was beyond the scope of this study. Therefore, the following section should be considered as an analysis of probable emitting infrastructure, *plus* possibly emitting co-located infrastructure.

Well pads were the most common type of oil and gas infrastructure sampled during our surveys (58% of total infrastructural emission sources). The infrastructure inventory we obtained from the provincial regulator identified several statuses of wells including Active, Abandoned, Cancelled, Completed, and Well Authorization Granted (WAG). It should be noted that Cancelled means that the permit for the well has been cancelled, usually before drilling has begun. Similarly, wells with the status of WAG may not have commenced drilling at the time we completed our surveys. However, based on discrepancies we noted in the field about abandoned infrastructure, we could not always rely on the accuracy of the status information in the inventory database. Furthermore, we assumed that test drilling and nearby infrastructure in these locations might serve as potential emission sources, so we chose to include wells with these status types in our analysis. A well with a Completed status means that the drilling was complete and the well was being prepped for production.

As noted earlier, we defined emission persistence in this study as the number of times a $CH_4$-rich plume was attributed to a piece of infrastructure, divided by the number of times we sampled that infrastructure in the downwind direction. We only attributed a plume to a piece of infrastructure if we recorded three or more successive $CH_4$-enriched measurements within 500 m in the downwind direction of the source. And in order for a piece of infrastructure to be classified as an emission source, it had to have > 50% emission persistence. Our technique of background subtraction is tuned to resolve small, localized plumes, but it should be noted that atmospheric conditions have a significant impact on the downwind detectability of emissions. In buoyant and unstable atmospheres, emission plumes will have a tendency to rise, and may not be detected reliably on the ground at distances of several hundreds of metres. As such, we would expect that the probability of detecting emissions on 100% of passes is lower than the probability of detecting emissions on 50% of passes. However, even a figure of 50% persistence (normally detected 2-3 times) indicates that there is high likelihood of a continuous emission at the site, though it might be of small scale, which is why we detect it only episodically. Many previous fugitive emission detection studies do not replicate surveys, but repeated emission detections help build both confidence in detection, as well as statistics about emission severity

and persistence through time. Operators and policymakers may find value in these data when prioritizing sites for further investigation, or mitigation.

Figure 6 presents the fractional emissions (emitting/surveyed) for each class of wells that we sampled on all six survey routes. We surveyed more Active wells than any other type, and their emission frequency was highest (47%). We sampled Abandoned wells second most, and their emission frequency was 26%. We sampled the remaining well classes less often, and their emission frequencies were 25% for Cancelled, 30% for Completed, and 27% for WAG.

While the frequency of emissions from well pads tended to be high, the concentration severity tended to be low. As noted earlier, no concentration above 8.148 ppm was recorded during the surveys themselves. Most of the anomalies were small-scale, and we detected them at roadside as $CH_4$ excursions on the order of ~0.1 ppm. While there might be appreciable inter-operator variability at the small scale, these sorts of statistics are expected because emissions are related to the type of infrastructure that sits in service, post-fracturing. This infrastructure is of course similar across the entire development, so it should not be surprising that well pads tapping the same formation 100 or 200 km apart might still have similar emission frequencies when the infrastructure of many operators are statistically bundled together. At the large scale, emission frequency might be an inherent property of the development, related to fluid type and handling, needed infrastructure, accessibility, and operator best practice.

A portion of the wells had operational statuses of Production, and the other portion was Undefined. Production wells were predictable emitters, with high statistical coherence from route to route (Fig. 7). We did not have a high enough sampling frequency of wells with other operation types (such as Injection, Disposal, and Observation wells) to calculate reliable emission frequencies so we excluded them from our analysis.

We sampled far fewer facilities than well pads, which was a result of the relative distribution of infrastructure types in this natural gas development. Overall, we found 32% of surveyed facilities were correlated with $CH_4$-rich plumes on > 50% of surveys. As shown in Figure 8, Compressor Stations emitted most frequently (70% emission frequency), which we expected based on the results of Omara et al. (2016). However, due to our low sample size relative to well pads, we would need to sample more Compressor Stations to arrive at a statistically significant estimate. Also, these larger facilities may emit from a height significantly higher above ground level than normal well pad infrastructure, which makes emission frequency measurements less reliable, and certainly conservative. In other developments where the road network allows for fuller transits around such stations at increasing distances, mobile surveying might be a good approach, but in the Montney, accessibility is often limited. In comparison to Compressor Stations, we were able to sample more Shared Facilities, Compressor Dehydrators, and Satellite Batteries, and we observed persistent emissions at a frequency between 11% and 28%.

Figures 6, 7, and 8 present only anomalies that were repeated on more than 50% of the passes when we were within the target radius of the infrastructure, and downwind. Figure 9 shows the emission persistence of each population of infrastructure type for all repeat surveys. As one moves to the right along the x-axis in Figure 9, emissions are more certain, less episodic, and likely also larger in magnitude - enabling more frequent detection across all atmospheric conditions. In the top left hand panel, it is clear that a group of about 60 out of 676 sampled Active wells were emitting persistently (100% of the times they were surveyed). In some cases, we detected these emissions on all six survey repeats on different days, and under different

weather conditions. As discussed earlier, it was predominantly the Active wells that emitted at 100% persistence, though several Abandoned and Cancelled wells were also highly persistent emitters. We detected emissions from the Undefined well category on an episodic basis. Of all fluid types, we detected the most persistent emissions from wells producing Gas, whereas we tagged Oil wells as emitters more episodically. The majority of facilities emitted at 50% persistence, although no facility

type dominated this trend. As can be seen from Figure 9, there is also an abundance of infrastructure that emitted at the 25% persistence level.

    Our results show that infrastructure type is a potential driver of emission patterns, which supports studies that have found large discrepancies in emission factors between valves used in different regions of the US (Allen et al., 2013). We did not have data on the types of equipment used at each well pad, but we did have information on ownership, operator size (via number of

sampled pieces of infrastructure), and well age (since spud date). In the Montney, the high number of newer wells emitted less frequently than the small number of older wells (Fig. 10). This is presumably because of improved modern practice, integrity, and better design of new valves, seals, and flange gaskets etc. There was a group of old infrastructure (> 50 years) in the Montney emitting with 100% frequency during our surveys. Infrastructure from larger operators tended to have lower emission frequencies, but this trend is anchored by a small number of small operators with 100% emission frequency at both 50%

and 100% emission persistence. It is important to note that many large operators grow through acquisition of infrastructure that previously belonged to smaller operators. As a consequence they will often inherit the environmental performance of companies whose assets they buy, and it may take some time to bring these sites in line with company expectations, which will skew our interpretations here.

    The bottom two plots in Figure 10 show severity of emissions (as measured by $e$CH$_4$ at roadside within the anomalies)

as a function of well age and operator size. These concentrations are shown "as recorded" and have not been corrected for dilution within the instrument cavity, and are therefore lower than they would have actually been if we were not in motion but stationary within the plume However, these figures still provide a useful relative index of emission severity. Overall, we see that the older infrastructure (> 50 years) has slightly elevated emission severity on-road. We did not note any clear relationship between emission severity and operator size.

As can be seen in Figure 11, there is no geographic trend to the emissions we detected throughout the Montney area; however, it is clear that certain areas, and potentially their associated infrastructure and practices, result in a higher number of emitting pieces of infrastructure (Fig. 11).

### 3.3 Minimum Detection Limit

Minimum Detection Limits (MDLs) allow emission detection studies to identify the measuring capabilities of the method being

used, and also to understand the minimum emission inventory within a development. Direct source measurement techniques often have lower MDLs than remote survey studies because the measurements are taken at the emission source over a longer period of time and often within a closed bag. For example, a study by Allen et al. (2013), that detected well pad emissions onsite, had an MDL of < 0.001 g/s. Not surprisingly, MDLs for truck-based surveys are lower, as noted in Brantley et al. (2014). In the study by Brantley et al. (2014), they came within an average distance of 57 m of the emission sources and collected data

for 10-20 minutes at each site of > 0.1 ppm $CH_4$. This translated to a MDL of approximately 0.01 g/s. In comparison, we were detecting emissions from farther away (319 m on average), and recorded gas concentration data for < 20 seconds at each site. However, our method of background subtraction and ratio-based plume identification allowed us to detect smaller concentration anomalies with confidence. Since concentrations will decrease away from a release source, small concentrations detected at distance could still represent moderately large emission severity. In order to estimate MDLs for this study, we established MDLs for various detection distances using cavity dilution experiments, followed by dispersion modelling.

Dilution in the instrument's measurement cavity is a function of anomaly duration (plume width, plus transit speed across plumes), and cavity size relative to pump rate. In a laboratory experiment we simulated dilution within the instrument using short injection pulses across a range of field conditions. We found that for realistic field conditions, the mean level of dilution was about 70%. In other words, the short pulses resulted in only 30% of the potential concentration deviation. Or, that observed concentrations were on average of 3.3 times lower than the actual ambient concentration that would be observed by a stationary analyzer. This dilution factor must be considered when interpreting our concentration readings at roadside, and also while calculating emission volume estimates. While it would possible to estimate an MDL for the hundreds of plumes separately, for simplicity we chose instead to focus here on mean MDLs.

Following the dilution experiments, we used the NOAA Air Resources Laboratory Gaussian Dispersion Model (*Draxler*, 1981) to determine the minimum $CH_4$ release rate that our mobile method distinguished from ambient at our various plume detection distances (minimum detection distance was 11 m, maximum was 496 m). One main assumption in the model is that the emission release occurred 1 m above ground level (AGL), however it is likely that we encountered varying emission source heights, particularly between wells and facilities. We also assumed the cloud cover to be 50% on all days, and that the cloud ceiling was an average height of 6096 m. The NOAA dispersion model computed the mixing depth using the wind speed, wind direction, and weather data we collected from our anemometer at 1 Hz sampling frequency throughout our surveys. Considering a dilution of 70%, and vertical and horizontal dispersion as simulated by the model under field conditions, we found that these conditions and plume concentrations corresponded to a minimum detection limit (MDL), or release rate, of 0.59 g/s at our average detection distance of 319 m. When we were very close to emission sources (< 60 m), we would have been able to detect emission rates as low as 0.065 g/s (with dilution considered). This exceeded the resolution of Brantley et al. (2014) at a similar distance, though in precision and not accuracy because the stationary techniques of Brantley et al. (2014) are designed to maximize volumetric estimation accuracy. The more precise MDL of our study is simply the consequence of being able to confidently resolve smaller concentration deviations from background using the ratio-based methods.

### 3.4 Methane Emission Inventory Estimate

Using MDLs for our study, we can reasonably estimate the minimum likely emissions inventory, because it is expected that infrastructural sources with larger emission rates cumulatively contribute the majority of $CH_4$ emissions (Frankenberg et al., 2016; Mitchell et al., 2015; Rella et al., 2015; Subramanian et al., 2015; Zavala-Araiza et al., 2015). According to a distribution of emissions at a US oil and gas site in the Four Corners region, emissions < 0.2 g/s did not significantly contribute to the

overall $CH_4$ flux rate (Frankenberg et al., 2016). If the US study by Frankenberg et al. (2016) reflects the emission patterns in the Montney, then our mobile method was able to capture the most significant emission sources in the area.

By applying calculated emission rates to the fraction of infrastructure we found to be persistently emitting, we estimated the total volume of $CH_4$ being released annually from sites emitting at rates above our MDL. Our emission frequency calculation
for Active wells (0.47) was very similar to the emission frequency of 0.53 that was recently calculated in the Alberta Montney near Grande Prairie (GreenPath Energy , 2017). Our method of calculating emission frequencies is corroborated by this recent FLIR study in the Alberta Montney, which increased our confidence in using our emission frequency calculations to estimate a minimum $CH_4$ inventory for the development. We used our MDL of 0.59 g/s to represent average emission rates from well pads in the Montney. This value is likely a conservative estimate because it is the smallest value detected at our mean detection
distance (319 m), and the majority of our emission detections occurred around this value (Fig. 3). It is also conservative because our method of attribution only considers the wells and facilities that were persistently associated with downwind plumes. It should be noted that this value overestimates emissions for the (small number of) well pads with detection distances < 60 m and emission rates < 0.59 g/s. However, (Brantley et al., 2014) showed that the largest sample population of well pads measured by OTM33A (n=107) had a mean emission rate exceeding 0.59 g/s. As a result, it is reasonable to assume that our MDL
serves as an average emission rate for well pads in a natural gas development, and one that allows us to estimate emission inventories for Montney well pads. For facilities, however, plumes are often emitted from higher above the ground surface, and the high concentration core of those plumes may not descend fully within a few hundred metres horizontal distance, to our 1 m AGL intake. As a result, the emissions we detected from facilities may significantly underestimate total emissions from those sources. For this reason, instead of actual measured MDLs we used previously-published natural gas facility emission volumes
of 2.2 g/s (Omara et al., 2016), combined with our emission frequency estimates for persistently emitting infrastructure, in order to estimate a total Montney-based source inventory.

The minimum reasonable inventory is given in Table 2. Based on the types of infrastructure we surveyed, and their corresponding 50% persistence emission frequencies, we estimate that total $CH_4$ emissions from wells are 8 216 tonnes per year, and total $CH_4$ emissions from the facilities we surveyed are 5 936 tonnes per year. We therefore estimate that, in total, there are
just over 14 150 tonnes per year of $CH_4$ emissions from all wells and facilities we surveyed in this study. If we extrapolate these values to cover all natural gas wells and facilities in the BC portion of the Montney formation (using infrastructure numbers derived from BCOGC GIS database), that translates to 72 900 tonnes $CH_4$ per year from wells, and about 39 000 tonnes $CH_4$ per year from facilities, totalling more than 111 800 tonnes $CH_4$ per year overall (3 564 000 tonnes per year $CO_2$e using a 100-year GWP of 30). These measurements and estimates represent emissions from infrastructure emitting > 0.59 g/s from our
average detection distance, and are therefore representative of the more significant, higher emitting sites in the area, and not small emissions that would be detectable only at close distance on the well pad. Furthermore, our estimates did not include some well types (including Cased and Drilled) for which our sample size was not large enough to reliably determine emission frequency, nor did it include transport-related emissions, or emissions from well completions. For these reasons, in addition to the measurement limitation imposed by our MDL, our calculations underestimate the actual $CH_4$ emissions from wells.
A comprehensive understanding of emissions in the BC Montney would also involve quantifying emissions below our MDL

(< 0.59 g/s), potentially using on-well pad screening surveys with our vehicle, and also onsite techniques to measure smaller emissions.

From all provincial energy sector practices, BC estimates fugitive $CH_4$ emissions to be 78 000 tonnes per year, and stationary combustion $CH_4$ emissions to be 17 000 tonnes per year (British Columbia Ministry of Environment (2012)). Our estimated

volume of 111 889 tonnes $CH_4$ per year (solely for infrastructure emitting > 0.59 g/s) suggests that Montney-related natural gas activity contributes more than 117% of this total value for BC. Our calculations are therefore higher than BC's emissions estimate when we consider that natural gas production from the Montney formation was 55% of BC's total production in 2014 (BC Oil and Gas Commission, 2014) (which would be equivalent to about 52 250 tonnes per year). It should be noted that the most recent available $CH_4$ emissions inventory from the province was from 2012, and that increased development and

production from the Montney since then may have increased what the regulator would expect to see from this development. However, the 2012 estimate was the most recent applicable emissions estimate we could locate to compare our estimate to.

Although our $CH_4$ emission estimates for the Montney exceed the estimates by the BC OGC, they remain lower than recent top-down oil and gas emission studies in the US. For example, in May 2014, Peischl et al. (2016) conducted airborne monitoring surveys of wells that produce more than 97% of North Dakota Bakken formation oil and gas and found that just

under 250 000 tonnes of $CH_4$ were being emitted annually. According to North Dakota state government records, there were 10 892 producing oil and gas wells in North Dakota at the time of the surveys by Peischl et al. (2016). This means that annual $CH_4$ emissions were an estimated $\sim$23.0 tonnes per well. Similarly, in 2013 Karion et al. (2015) performed airborne surveys over the Barnett shale in Texas and estimated that just over 525 000 tonnes of $CH_4$ are released annually from this development. Texas state records show that as of early 2013 there were 16 821 producing oil and gas wells accessing the Barnett shale

formation, which means that annual $CH_4$ emissions in this development were $\sim$31.3 tonnes per well. The analogous figure in the Montney is $\sim$7.3 tonnes per well based on our volume estimate. The lower emission frequencies per well in the BC Montney are consistent with the relatively low occurrence of excess atmospheric $CH_4$ in the region on all surveys compared to higher atmospheric $CH_4$ values recorded in US developments. Although airborne measurement techniques are not ideal for locating exact emission sources, they are well suited to calculate total emission volumes for entire regions so long as other

emission sources (such as agricultural) can be accounted for, which they were in the studies listed above. The top-down nature of mobile surveys for large amounts of infrastructure allows for a comparison between our $CH_4$ volume estimate and those of Peischl et al. (2016) and Karion et al. (2013).

### 3.5 Uncertainty in Plume Detection, Attribution, and Volume Estimation

There are several sources of uncertainty in a study involving plume detection, geospatial attribution of plumes, and annual vol-

ume estimates. The sources of uncertainty act individually, and in combination. We performed survey repetitions and included Control Route experiments in the study design to help minimize uncertainty.

We calculated plume detection uncertainty by surveying our pre-planned Control Route three times and calculating the incidence of false positives from the collected gas concentration data. This allowed us to quantify our probability of falsely detecting a $CH_4$-enriched plume from other possible sources in the area. The Control Route was laid out on the periphery of the

development, and presented similar sources of landscape emissions. It also included additional types of industrial activity that may have been sources of atmospheric $CH_4$ (such as a pulp mill and active logging), which were not present on the other routes. The Control Route had a small amount of oil and gas infrastructure, but for our uncertainty analysis we excluded all datapoints within 5 km of those sources. We used the gas concentrations collected on all three replicate surveys of the 370 km Control

Route to calculate the fraction of datapoints that our method falsely interpreted to be part of a plume. The Control Route false detection probability was 0.2%, but the false positives did not re-occur in the same locations and consisted of mostly random short-term deviations. Since all our oil and gas infrastructure surveys were repeated 3-6 times, and our oil and gas detection criteria required multiple independent and confirmatory plume detections at the same location (generally on different days), the probability of falsely detecting a persistent emitter was extremely low (< 0.01%). False plume detection was therefore unlikely

in this environment, and not a significant contributor to overall uncertainty. In other words, when plumes were detected, we are highly confident that one or more oil or gas related sources were nearby.

Compared to plume detection, there is more uncertainty attached to the process of attributing on-road $CH_4$ plumes to specific pieces of oil and gas infrastructure. Firstly, there is some uncertainty in the infrastructure database from the provincial regulator, because such records are rarely without error or perfectly up to date. Ground-truthing all locations in the provincial regulator's

infrastructure database was not feasible and therefore pieces of infrastructure might be mischaracterized, or even non-existent, whereas in other cases we may have passed wells or facilities not yet in the database. However, we assume that the regulator's online database was mostly up to date. Additionally, widely dispersed plumes in areas of high infrastructure density may increase uncertainty in our attribution method. For example, in some cases, multiple well pads and/or facilities may have been within 500 m in the upwind direction of a plume, and so we may have mischaracterized specific pieces of infrastructure as

emission sources. To define the likelihood of this happening, we performed an analysis of persistently tagged sites that could not reasonably be emitting (i.e. wells with a status of "Cancelled"). This analysis showed that >95% of these sites had other potential upwind sources at 500-1500 m distance (which is still a reasonable distance for plume detection given emissions of sufficient magnitude). So, with this method alone, the potential for tagging the wrong infrastructure as emitting is reasonably high. However, in our study we sampled downwind of the same sources 3-6 times, and generally under a variety of wind

directions, and so the replicated surveys in our study helped mitigate this issue. A large population of wind directions, combined with back-trajectory analysis, as well as emission persistence criteria, in many circumstances will have helped identify which piece of infrastructure (of several co-located nearby pieces of infrastructure) was the most probable emission source. Although attribution uncertainty affects our understanding of contributing infrastructure (type, age, owner), it does not severely impact either emission frequencies, or volumetric inventory estimates. For example, since we are very certain about plume detections,

we can also thus be confident in our total emission frequencies (total plumes versus total pieces of infrastructure) for the development, even without knowing the precise origin of each plume with perfect certainty.

To determine uncertainty in our volume estimates, we used Gaussian plume dispersion analysis to estimate MDLs from all well pads individually. These calculations took into account the smallest measurement of frequently detected $CH_4$ departures from background, as well as the individual distances between plume and source. To bracket our estimated $CH_4$ inventory, we

used one standard deviation from these MDLs as an estimate of total uncertainty in our volume estimate. The standard deviation

of the MDLs from all plumes that were attributed to well pads was 0.14 g/s. Propagated through the inventory calculations, our uncertainty in the $CH_4$ inventory for the Montney development is +/- 15 700 tonnes per year. It should be noted that our inventory estimates are based strictly on MDL exceedances, and it is therefore likely that the actual inventory might surpass 111 800 +- 15 700 tonnes $CH_4$ per year.

5    Overall, when we detect a plume on-road, we are confident that it is from oil and gas related source. We are less confident in plume attribution accuracy in areas of dense infrastructure. However, because we know the plume exists, we can be quite certain of total emission frequency values (emitting/surveyed). Our MDL-based inventory approach is suitable for this development where few large plumes were observed, and provides a conservative, but still useful, $CH_4$ inventory estimate.

## 4    Conclusion

10    Unconventional natural gas development in the BC Montney began less than a decade ago, and so the majority of infrastructure is new in comparison to many old conventional oil developments in Alberta and Saskatchewan. Though the Montney is regarded as a young development, there are many locations where old, decommissioned infrastructure exists, and in a generally unkept state. Our results show that older infrastructure is more prone to persistent leaks, albeit at similarly low $eCH_4$ severity in comparison to younger wells. These results reinforce the need for regulators to pay attention not only to modern equipment, 15    but also legacy wells and infrastructure.

In calculating the frequency of emissions in the BC Montney above our MDL of 0.59 g/s, we found that about 47% of Active wells were emitting. Abandoned wells were also associated with emissions at 26% of the 228 sites we sampled, and we located a group of aging infrastructure (> 50 years old) that was emitting every time we sampled downwind. The emissions we detected from facilities were consistent in both presence and $eCH_4$ severity, however our mobile detection method is sensitive 20    to plume transport turbulence associated with emissions higher above ground level such as flare stacks.

Our calculated emission frequency values, combined with estimated and pre-established emission factors for wells and facilities, provided a $CH_4$ emission volume estimate of more than 111 800 $\pm$ 15 7000 tonnes per year for the BC portion of the Montney. This value exceeds the province-wide estimate provided by the government of British Columbia even though the Montney only represents about 55% of BC's total natural gas production. But, in comparison to studies at select US natural 25    gas sites (Peischl et al., 2016; Karion et al., 2015), our results suggest that natural gas activity in the Montney formation may emit both less frequently and less severely than US comparators.

Methane emission reduction strategies for large natural gas developments such as the Montney should focus on first locating super-emitting sites, and then following up with site-specific emission techniques such as FLIR cameras. This strategy would support LDAR already in place, in a way that would minimize cost to individual operators. It would also focus the attention on 30    the problematic infrastructure and operators, and does not share the cost burden across companies that have already invested heavily in emission reduction technology and leading best practice. It is feasible to detect super-emitters through exhaustive survey campaigns, even from roadside campaigns such as this one. Our results show that a mobile surveying approach for large developments such as the Montney can help to locate probable emitting infrastructure pieces that contribute to the heavy-tailed

emission distribution found by Frankenberg et al. (2016). Also, using a mobile survey method to define persistently emitting infrastructure allows for the emission type (consistent or episodic) to be deduced. Our study highlights the need for emission reduction efforts in the Montney to be focused on the few higher-emitting Active Gas wells, as well as Abandoned, and aging infrastructure.

5 ## 5 Data availability

Datasets of atmospheric gas concentrations, wind, and temperature data are available upon request. Oil and gas infrastructure location data can be accessed through the BC Oil and Gas Commission Open Data Portal (BC Oil and Gas Commission, n.d.)

*Author contributions.* D. Risk and M. Lavoie developed the algorithms for background subtraction and plume detection. E. Atherton designed the field campaigns with insight from J. Werring. E. Atherton, J. Werring, A. Marshall, J.P. Williams, and C. Minions carried out
10 the field surveys. The data were compiled and analyzed by E. Atherton with help from M. Lavoie and C. Fougere. E. Atherton and D. Risk prepared the manuscript.

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

**Table 1.** Survey statistics by Route. Route locations are shown in Figure 1.

| Routes | Control | Regional | 1 | 2 | 3 | 4 | All |
|---|---|---|---|---|---|---|---|
| Route Length (km) | 370 | 545 | 145 | 210 | 235 | 280 | 1785 |
| Number of Repeat Surveys | 3 | 3 | 6 | 6 | 6 | 6 | 30 |
| Total km Surveyed | 1110 | 1635 | 870 | 1260 | 1410 | 1680 | 7965 |
| Unique Sampled Wells | 152 | 436 | 172 | 241 | 298 | 182 | 1481 |
| Unique Sampled Facilities | 10 | 113 | 63 | 29 | 34 | 16 | 265 |
| Unique Sampled Groups | 49 | 304 | 146 | 88 | 110 | 51 | 748 |

**Table 2.** Emission volume calculations for all surveyed infrastructure, and also extrapolated to account for all wells and facilities within the BC portion of the Montney formation. Our minimum detection limit (MDL) of 0.59 g/s was used as the emission factor for wells. Facility emission volumes are from Omara et al. (2016) because our sampling from facilities was probabilistic due to emission height variance.

| Type | Infrastructure n | Emission Freq (%) | Emission Volume (tonnes/year) | Emission Total (tonnes/year) |
|---|---|---|---|---|
| **Surveyed Wells** | | | | |
| Active | 676 | 47 | 18.6 | 5910 |
| Abandoned | 228 | 26 | 18.6 | 1103 |
| Cancelled | 130 | 35 | 18.6 | 846 |
| Completed | 64 | 30 | 18.6 | 357 |
| Surveyed Facilities | 265 | 32 | 70 | 5936 |
| Total $CH_4$ volume | | | | 14152 |
| | | | | |
| **Montney Wells** | | | | |
| Active | 5294 | 47 | 18.6 | 46280 |
| Abandoned | 2149 | 26 | 18.6 | 10392 |
| Cancelled | 1989 | 35 | 18.6 | 12948 |
| Completed | 582 | 30 | 18.6 | 3248 |
| Montney Facilities | 1742 | 32 | 70 | 39021 |
| Total $CH_4$ volume | | | | 111889 |

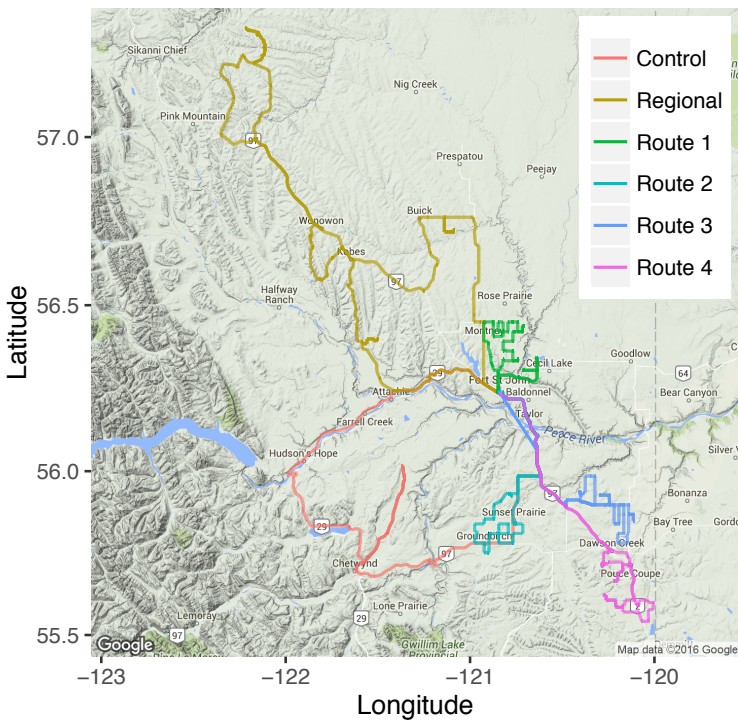

**Figure 1.** Map of mobile surveying routes. Each route was surveyed six times in August - September, 2015. The Regional Route and Routes 2-4 dissected unconventional natural gas developments. Route 1 surveyed conventional oil. The Control Route was located in an area with a comparatively small amount of oil and gas development, although due to lack of accessible roads in the area it passed by some infrastructure on Route 2 upon returning to the Fort St. John area.

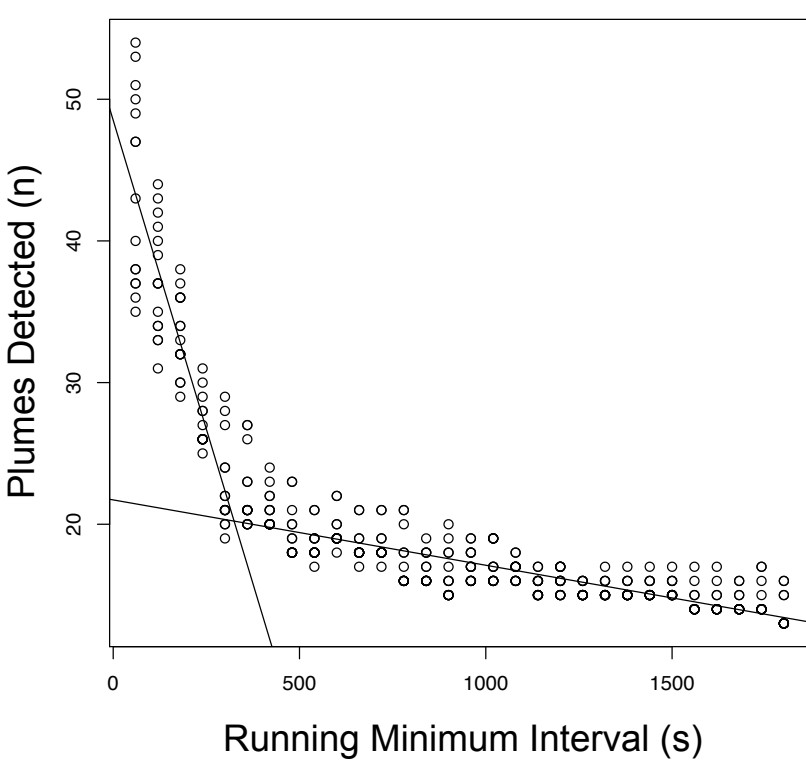

**Figure 2.** Example of a regression plot that demonstrates the optimization process we used to calculate an RMRI for each survey. The RMRI for each survey was chosen where the two linear regression lines intersect.

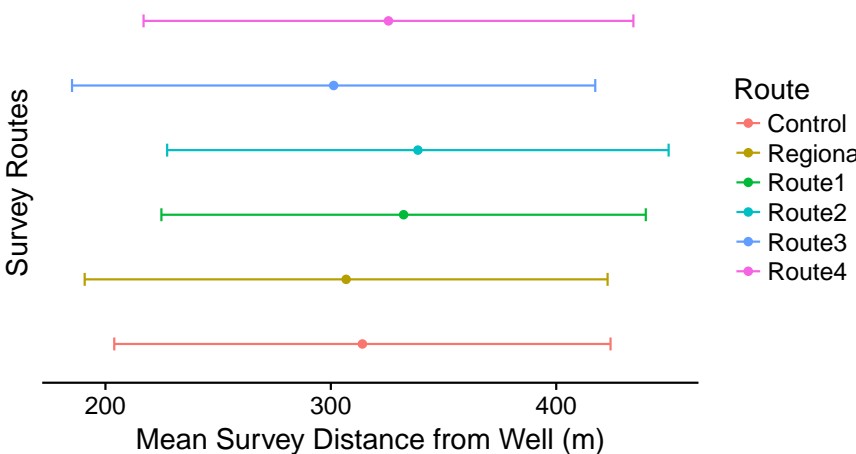

**Figure 3.** Mean distance from infrastructure while surveying each of the six routes listed in Figure 1. One standard deviation from the mean shows the range of distances at which we were sampling downwind of infrastructure.

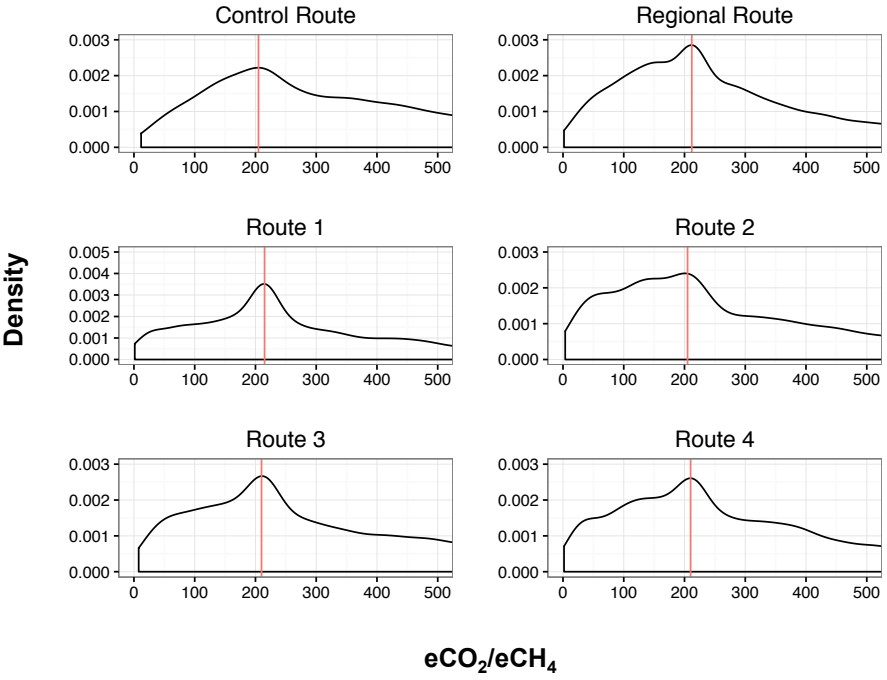

**Figure 4.** Kernel density plots showing the density of $e\mathrm{CO_2}{:}e\mathrm{CH_4}$ measurements on each route. Red vertical lines indicate natural $e\mathrm{CO_2}{:}e\mathrm{CH_4}$ values about 215. Methane-enriched peaks are visible to the left of the natural ratio on all routes except for the Control, where the slope approaches zero with no peaks because substantially less natural gas infrastructure was surveyed. Ratios higher than the natural represent $\mathrm{CO_2}$-rich plumes which would not be caused by natural gas related emissions, but likely diluted car exhaust fumes, or other industry types.

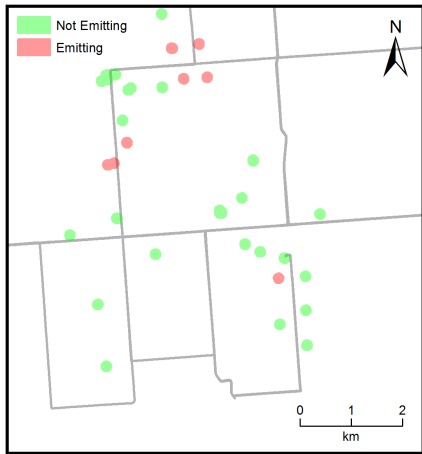

**Figure 5.** A subset of infrastructure locations that we surveyed during our field campaign in attributed form. This figure serves as an example of how we attributed wells and processing facilities to on-road plumes. Grey lines represent the survey route. In this case 31 wells or facilities were surveyed, and we used our attribution technique, which accounts for wind direction and distance to source, to determine whether or not these wells and processing facilities were probable emission sources.

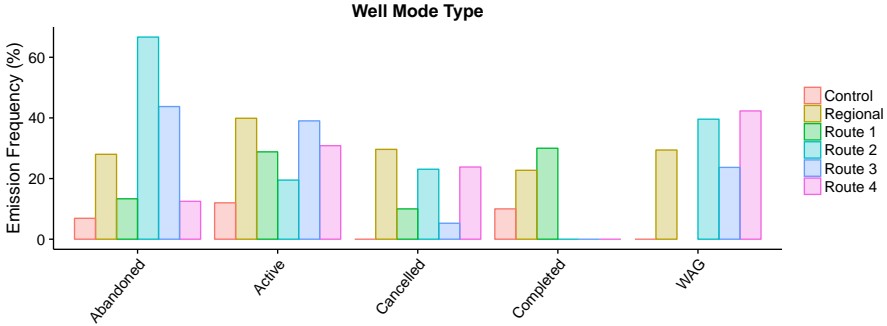

**Figure 6.** Emission frequencies for each well mode type for all surveyed infrastructure on each route. These emission frequencies were considered in our total emissions inventory calculations.

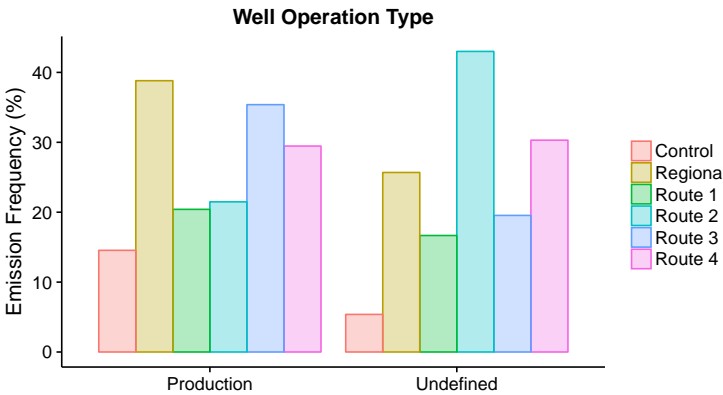

**Figure 7.** Emission frequencies for each well operation type for all surveyed infrastructure on each route. Certain operation types for which we did not have a representative samples are not included (such as Injection and Disposal wells).

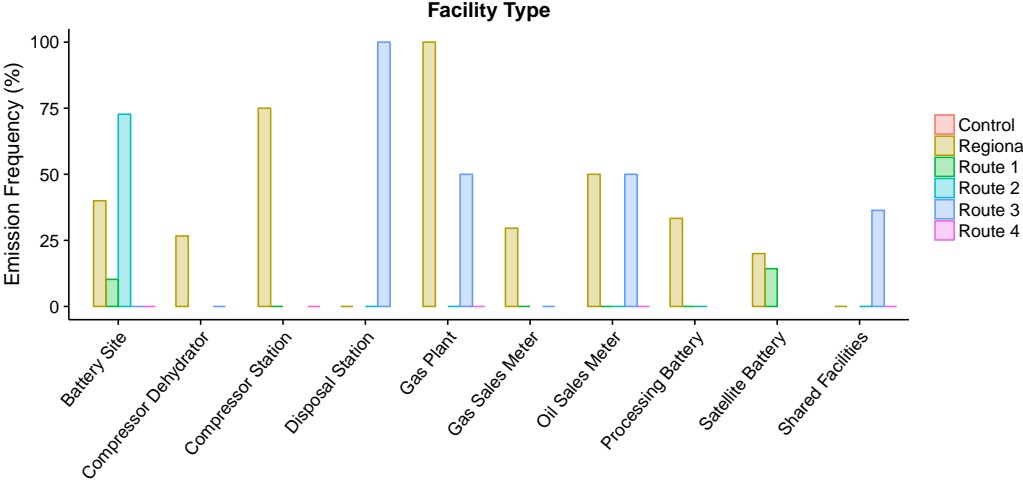

**Figure 8.** Emission frequencies for each facility type for all surveyed infrastructure on each route. These emission frequencies were considered in our total emission inventory calculations.

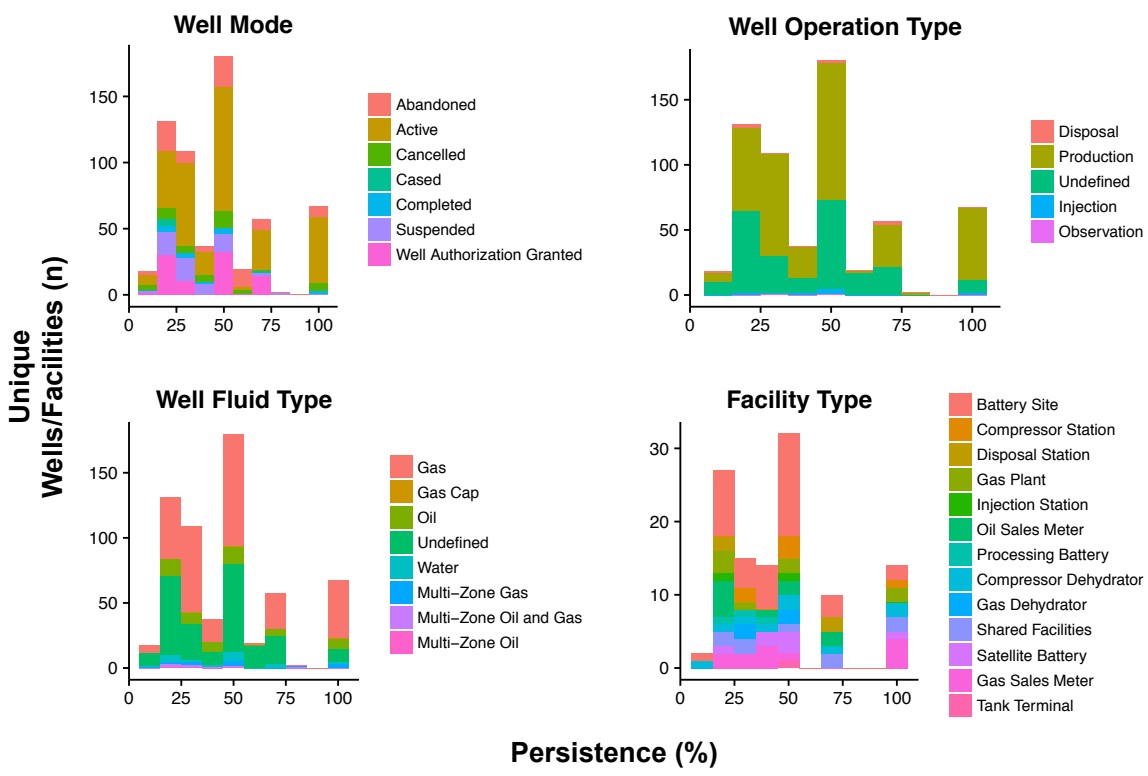

**Figure 9.** The cumulative number of unique wells/facilities versus emission persistence (%) across all 30 mobile surveys. Persistence refers to the repeated tagging of a piece of infrastructure as a possible emission source based on the method of plume attribution we applied in this study.

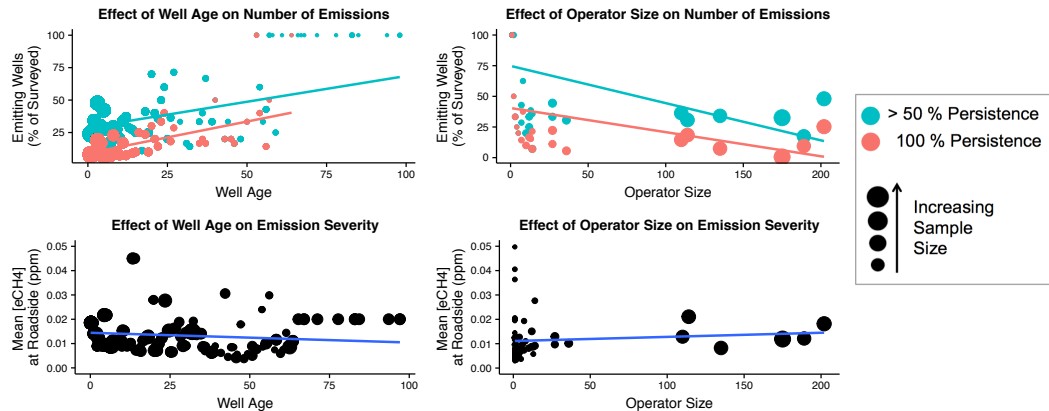

**Figure 10.** Effect of infrastructure age and operator size on detected emissions. The sizes of the dots represents the number of samples taken. Red dots are those recorded at the 100% persistence level, green dots are at 50% persistence.

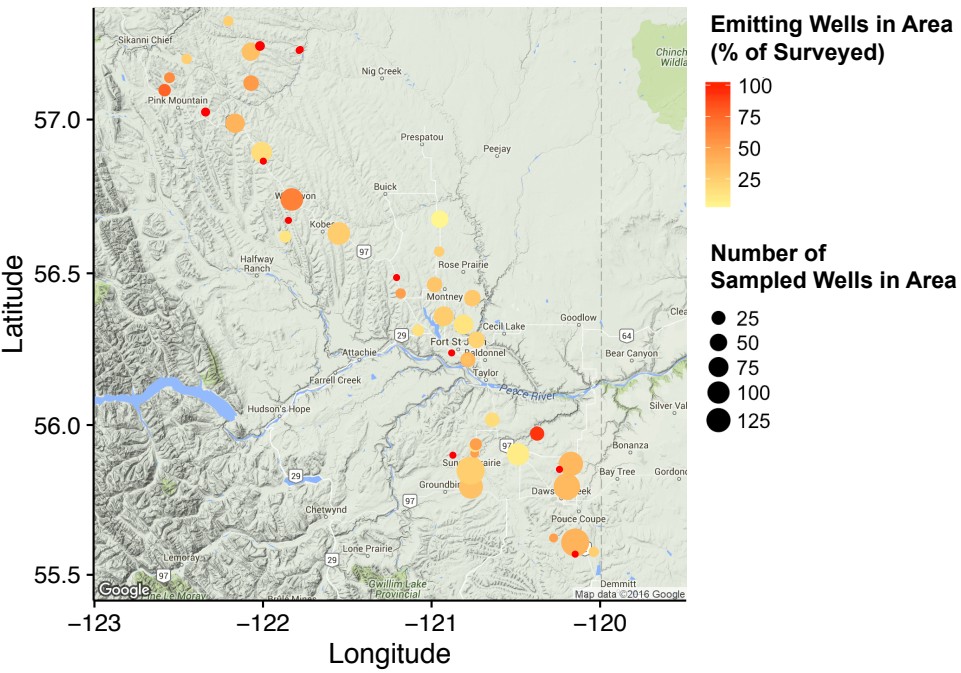

**Figure 11.** Distribution of emitting infrastructure by industry-defined area. The size of the circles represents the number of measurements we took downwind from individual wells or facilities in each area. The colour of the circles represents the frequency of emitting infrastructure in each area.