# Peer review of "Mobile measurement of methane emissions from natural gas developments in Northeastern British Columbia, Canada"

_Atmospheric Chemistry and Physics, 2017_

## Referee Comment (RC1) · Anonymous Referee #2 · 10 May 2017

General Comments

* The manuscript is extremely well written. * This paper addresses an important need in the community with a practical and well-described method for estimating emissions rapidly and on a broad scale. * While I understand that there was not an opportunity to benchmark the estimates against other methods of emissions estimation, the lack of validation remains a significant weakness. I nevertheless recommend publication, but this caveat should be recognized at key steps in the analysis. * The largest omission from the paper is the lack of any uncertainty estimate for the emissions from the region. Some effort should be made to rectify this in the final paper. * I don't understand the use of linear regressions (with variable slope and offset) for the detection rate estimates.

Justification of why this analysis should be used over the simple calculation of rate = emitting sources / total sources should be provided, or the authors should revert to the simpler analysis.

Specific Comments: - P1 L17: emissions estimates for the Montney development does not have an uncertainty estimate. It is difficult to interpret the emission results without an uncertainty associated with it.

- P5 L1 - 10: The authors state that they are using excursions in the eCO2:eCH4 ratio (<150) as indications of natural gas emissions. However, I would imagine that other sources of CO2 could add noise to this ratio (especially since there are other vehicles that contribute to excess CO2). Figure 3 further indicates this issue. A fairly obvious alternative would be to use the same RMRI algorithm and use eCH4 > threshold as a criterion for when emissions are detected. It would be helpful if the authors could provide some more justification why the ratio eCO2:eCH4 is a better metric than simply eCH4.

- P5 L10-12: "Our optimal RMRI was taken to be the point at which anomalies were maximized, but also where we avoided the rapid noise-associated increase associated with extremely short RMRIs": in practice, how was this optimization performed? It appears to be a subjective choice. Is this true? It would be preferable if the choice was made objectively using quantitative criteria; it would also be preferable to have the same algorithm be used for all surveys.

- P5 L 18-19: "Combustion values were also recorded along the routes when eCO2:eCH4 exceeded 1000, and were related to vehicle tail-pipe emissions and in-dustry". What does 'combustion values' mean?

- P5 L24-25: "because ratios are more conservative than concentrations in valleys and other areas where pooling of 25 gases is common, and fewer false positives are likely" - doesn't the RMRI algorithm take care of slowly varying concentrations of CH4? It would be good to demonstrate clearly why eCO2:eCH4 is an advantage; if one were to

reproduce this method at a larger scale, it would be good to provide clear understanding of why the CO2 concentration is required.

- P5 L28-30: why was the value 150 selected? What is the effect of this selection on, for example, the emissions estimate, the number of emitters detected, the detection limit, etc. Similarly, what is the effective limit on detection of the system, in units of eCO2:eCH4?

- P6 L7: are there any estimates of cattle emission in this region that could be included?

- P7 L10: how is this probability defined? Per mile? Per second? For the whole route? This isn't clear.

- P7 L1-5: The kernel density plots do not have a clear knee below 215. Where is 150 on this plot? why was 150 selelected, and not 125 or 175, for example?

P7 L16-20 and Fig 4. Was wind direction used to evaluate whether a plume should have been detected from the green well pads? Are the databases of well locations up to date? Was there an effort to corroborate locations with on-ground survey or satellite imagery?

P7 L32: "it had to have > 50% emission persistence." Similarly, did persistence include wind direction? In other words, did persistence include whether the potential source was upwind of the vehicle at the moments the vehicle passed by?

P11 L8: "concentrations will decrease exponentially away from a release source": the dependence on distance is not exponential. Gaussian plume models predict something like $\sim 1/d$ to $1/d^2$, for example.

P11 L11-18: Wouldn't nearby plumes (with faster time signatures) be diluted more than more distant plumes? And wouldn't the peak area (in time) be conserved for short pulses? This is a very big adjustment of the concentrations and therefore the emissions. Did you use peak height or peak area to estimate emissions?

P12 L9: Rather than using the MDL as the average estimate of emissions, wouldn't it be possible to actually craft an estimate of emissions given the plume dispersion model and estimated distances?

p12 L28: It is important to include some uncertainty estimates for the emissions estimate. Even a simple low and high estimate of error is better than nothing. For example, the estimates of errors on the slope of the active wells could be used to bound the estimate.

p14 L9: It's not clear how this method identifies super emitters, since the authors do not present a clear method for quantifying emissions and identifying the largest emitters. How does this method help identify the largest emitters?

Fig 5: In some panels (e.g., the top panels), the regression lines do not pass through zero. This doesn't make any physical sense. Why should there be a threshold for number of wells surveyed below which no emissions should occur? Why would there be no emissions for surveys with fewer than 60 wells surveyed? I don't understand the rationale for a linear regression. Why not simply ratio the total number of sites with emissions / total number of sites surveyed across all surveys for each category? This would make more intuitive sense. Alternatively, the linear regressions could be forced through zero, which would be similar.

Fig 6 and 7: similar comments to above for Fig. 5.

Fig 8: Is the occurrance structure due to the fact that some areas were surveyed only three times, which did not allow for a 50% persistence point, for example? This set of plots is a bit confusing.

Fig 9: what do negative mean eCH4 excursions mean (gray bars of lower panels)?

Fig 10: could you add in the survey paths on this plot for reference?

Typographical error and other small comments P1 L13-15: "older infrastructure tended to emit more often (per unit) with comparable severity in terms of measured excess

concentrations on-road." - unclear; per unit? what is a unit? reword for clarity, please

---

## Referee Comment (RC2) · Anonymous Referee #3 · 11 May 2017

The authors present data and analysis from six mobile measurement surveys in the Montney formation which include methane emission concentration and rate information from 1600 passes near wells. The routes were surveyed 3-6 times each and designated as new wells, old wells, and a control. The authors use the methane and $CO_2$ concentration and meteorology data to calculate emission rates of methane from wells. They analyze the data using online well number, production, age, etc. information to show which types of wells or activities emit most or most often. And finally, they compare their results to available data from recent studies in other formations in U.S. Collection of mobile data, especially when one is at the whim of wind to assure downwind of well measurements, is no easy task. The authors have conducted a great

survey of sites in the Montney formation. This study is exactly the type of research that is needed to clarify and quantify the emission rates of methane from different formations and sources. The authors have done a lot of work and the publication of this paper (especially with the availability of the data upon request, as noted at the end of the manuscript) will be a great addition to the current body of knowledge on methane emissions from oil and gas sources. However, there is some more analysis, organization, and sentence structure improvement that is needed for this paper before publication. Please see my General and Specific comments below: General Comments: 1. Various groups have used different approaches to quantifying methane emission rates (e.g., EPA's OTM 33 method, use of different tracers for close or far quantifications using the Tracer Ratio Method, reverse plume modeling, etc.). One of the things that all the methods above have in common is method validation. It seems that the authors of this paper have not conducted any method validation studies. This is a major weakness in the study. I would recommend that a quick methane and $CO_2$ release study and measurement be added to the paper. However, I understand that time and funding may not be available to do this. Instead, I suggest the authors do a detailed uncertainty analysis (maybe even add a section to the paper) where they discuss and calculate a theoretical uncertainty for their measurements and calculations. The authors have a short section on this, but since no method validation has been done, the uncertainty analysis should more exhaustive. 2. Another point that is not clarified in this paper is the difference between measurements made from unconventional vs conventional wells. The authors make a distinction between new and old wells. The attribute the increase in the oil and gas activity in the area to the use of unconventional extraction methods. However, when they discuss the wells measured, they do not show any information on the unconventional vs conventional wells. Are all the wells measured unconventional? 3. The authors do not distinguish between short term operations and permeant emission sources in their calculations. This may be difficult to do, but at least a discussion of how these would affect the regional emission calculations should be added. 4. Some of the writing in the paper is confusing. The sentence structures do

not flow well. I have given some specific examples of this in the "Specific Comments" section, but strongly suggest the co-authors who were not directly involved in the writing of the manuscript read the paper and comment on sections. Sometimes it is easy for the authors to unintentionally disregard clarity as they themselves are so familiar with the subject of the study. 5. The authors use two different tenses and two different voices (active and passive) throughout the paper. I suggest choosing only one. Two different voices and tenses make it confusing for the reader and require re-reading of sections. Specific Comments: 1. Abstract: The writing style of the abstract does not lend itself to clarity. The flow of the sentences is not coherent. I suggest re-writing it for better clarity and flow. For example: "We also observed emissions from facilities of various types that were highly repeatable." is one of the sentences that is unclear and confusing. Or "This value exceed reported bottom-up estimates of 78,000 tonnes for all oil and gas sector sources in British Columbia, of which the Montney represents about 55% of production". The abstract starts very abruptly. I suggest rewording the first sentence. 2. Page 1, Line 2: What do the authors mean by "incidence"? 3. Page 1, Line 4: Are authors including all oil and gas locations in "development". I suggest clarifying this or using a different word. 4. Page 1, Line 6: The use of "infrastructural" here has the same problem as the previous comment. 5. Page 2, Line 5: What do the authors mean by "a petroleum system". 6. Page 2, Line 14: Please rewrite "Over a 100-year... (ICPP, 2014)" for clarity. 7. Page 2, Line 33: I have noted this in the abstract too. Please describe what you call "infrastructure". 8. Page 3, Line 1,2: Please re-write sentence for correct grammar. 9. Page 3, Line 13: Please define "super-emitters" and use appropriate references. 10. Page 3, Line 26: Do the authors have some estimate of numbers of wells? 11. Page 4: The authors use the words unconventional and hydraulically fractured interchangeably. These two do not mean the same thing. Unconventional oil and natural gas extraction refers to both hydraulic fracturing and horizontal drilling. 12. Page 4, Line 8: Is 1Hz frequency the rate of data collection? 13. Page 4: What were the average distances from wells? If this data is available, can it be used with meteorology data for plume dispersion modeling? 14.

Page 4, Line 14: Please re-write for correct grammar. 15. Page 4: Please note which routes the numbers are based on in Figure 1. 16. Page 4, Line 19: What do the authors mean by "raw" ? 17. Page 4, Line 23: What are wind speed units? 18. Page 4, Line 25: Since the authors have given the manufacturer of the other instruments used, why not indicate what type of GPS was used? 19. Page 4, Line 32: Please re-write "However, our surveys. . . unusable." for clarity. 20. Page 5: Were the same approaches used for both CO2 and CH4 data handling and analysis? Please add a few sentences to clarify this. 21. Page 5, Line 10-12: Please add some statistical data. 22. Page 5, Line 20-21: Please re-write for clarity. 23. Page 5, Line 21: What do the authors mean by "normal air"? 24. Page 5: What are some sources of CO2 in the area? As this can be a major concern in your calculations, please add a few sentences to address this. 25. Page 6, Lines 1-2: Please re-write for clarity. 26. Page 6, Line 2: What do the authors mean by "developmental"? 27. Page 6: Are there any large dairy operations in the area? 28. Page 6, Line 15: Please re-write sentence for clarity. 29. Page 6, Line 16: I thought the authors used one route as control. Did they actually make measurements from oil and gas structures on this route and include them in the analysis? If yes, then should the designation not be changed? 30. Page 6, Line 19: Following up on the previous comment, please give numbers of the differences in the oil and gas densities. 31. Page 6, Line 30: What was the speed of the car during these measurements? This is important as it can have an impact based on the width of the plumes. 32. Page 6, Lines 31-32: What is the difference between 314 and 319 meter designations? Also, should this not be in the methods section instead of the results section? 33. Page 7, Line 1: What do the authors mean by " In each, we see a peak of signatures near ∼215 which is representative of natural"? 34. Page 7, Line 5: relative to what? 35. Page 7, Line 25: What are the other methods? 36. Page 7, Line 27: Should this be associated? 37. Page 7, Line 30: Please define what you mean by ". . . a piece of infrastructure. . ." 38. Page 7, Line 32-34: Please re-write for clarity. 39. Page 7: I suggest adding clarifying sentences like, Well pads were the most common oil and gas structures encountered/sampled during our survey (%# of total sites).

40. Page 8, Lines 5-8: Please re-write for clarity. 41. Page 8, Lines 14-17: Please use a consistent theme for capitalization. 42. Page 8, Line 20: Please replace the term "probably" with one with a more scientific connotation or even some statistics. 43. Page 8: Please explain, clearly, what each category of wells encompasses. For example, does authorization mean that permit was granted? Was work on the pad started? Was temporary drilling part of the study or as noted previously was it excluded? 44. Page 9, Line 27: 60 out of how many? 45. Page 10, Line 2: Please reword "... less emission prone..." 46. Page 10, Line 20-32: This paragraph does not belong in this section. I suggest either deleting it or moving it to a more appropriate location. 47. Page 12: Please give a more detailed (method definition, details, and statistics) of the setup of your calculations. 48. Page 13, Line 5: Have the number of wells changed since 2012? Would this affect the calculations in this paper, especially when dealing with the comparison to other sites/studies? 49. Page 13: Please add a discussion of possible reasons for the differences in this study and others noted here. Uncertainty range? Different basins? Different measurement approaches? 50. Page 13, Line 29: Please give numbers. 51. Please revise the Conclusion. It needs more specific numbers and information. Also, the addition of super-emitters at the end does not make sense as this paper was not directly making measurements from such sites based on the previously discussed statistics. 52. Figure 1: Is it possible to add the location of the wells here as a light gray background? It would be helpful in visualizing the type of routes. Also, please make sure that your designations of routes in this figure and the paper are the same. After reading through, I found TABLE 1 in Tables. Do authors mention this table in the text of the manuscript? 53. Figure 2: What are 88 industry-defined areas? 54. Figure 3: This is not a comment on this figure, but in looking at this and other figures, having a table with route numbers, names, and characteristics would be very helpful. Something like Table 1 55. Figure 4: Please revise caption to explain graph better. What are the gray lines? 56. Figure 5: Please re-write caption for clarity. Also, the addition of the uncertainty discussion as noted before, will help this figure. 57. Figure 7: Why are there zero-zero points in this graph? Although physically

a zero-zero point makes sense, I do not think the addition of the points is statistically sound. 58. Figure 9: Please add numbers in the increasing sample size legend. Were any of the wells in this area re-worked? This will change the definition of well age in this discussion.

---

## Referee Comment (RC3) · Anonymous Referee #1 · 11 May 2017

**Review of Atherton et al. for Atmospheric Chemistry and Physics**

***General Comments***

  In "Mobile measurement of methane emissions from natural gas developments in Northeastern British Columbia, Canada", Atherton et al. describe with lucidity and apply with care an improved mobile survey technique for identifying methane leaks in an understudied region of Canada's oil and gas fields. The measurements are used to probe which aspects of the oil and gas infrastructure in the portion of the Montney region surveyed are most likely to emit methane. A conservative estimate of the bottom up inventory for entire Montney development is calculated and compared against state-based estimates, which is the most uncertain part of the analysis. The manuscript clearly describes the measurement and analysis techniques, highlights the limitations of the approach, and contextualizes the results nicely. I recommend this manuscript for publication in Atmospheric Chemistry and Physics with only minor changes.

***Specific Comments***

| Line | Comment |
|---|---|
| p. 2, l. 13 | "ostensibly less environmental impact" – People have been more concerned about water-based impacts of hydraulic fracturing than those of coal, so restating this perceived advantage to be specific to atmospheric drivers of climate might be more accurate. |
| p. 3, l. 13 | "super-emitters, and reduction" should be "super-emitters and reduction" |
| p. 3, l. 26 | "significantly, with thousands" should be "significantly with thousands" |
| p. 4, l. 3 | "August 14 2015 and September 05 2015 we" should be "August 14, 2015, and September 5, 2015, we" |
| p. 8, l. 20 | "probably" should be "probable" |
| p. 9, l. 7-8 | Indeed, accurate infrastructure inventories can be difficult to maintain. This statement seems to indicate that the correlations were not what was expected, which led to suspicion of the infrastructure inventories. Could you rephrase this statement to describe the limitations on analysis that uncertainties in the inventory induce? |
| p. 7, l. 25 | "FLIR" is first used here, but the acronym is first defined on page 10. Could you please reorder? |
| p. 12., l. 20-1 | "Montney based" should be "Montney-based" |
| Figure 2 | If I understood correctly, industrial sources were passed on multiple routes. Could these dots and bars be color-coded (with colors from Figure 1) by the route on which the source was observed? |
| Figures 5,6,7 | Please add to the caption the meaning of the grey-shaded areas around the line. |

---

## Short Comment (SC1) · 30 May 2017

Comment 1: -Page 12 Line 20 refers to Omara et al and quotes a "natural gas facility emission volumes of 2.2 g/s".

-Reading through the Omara paper it is not immediately clear where this value originates. As per the Omara abstract:
"mean facility-level CH4 emission rate among UNG well pad sites in routine production (18.8 kg/h (95% confidence interval (CI) on the mean of 12.0-26.8 kg/h))"

-Note that 18.8 kg / h works out to 5.2 g/s.

[Figure]

-Clouding the issue is a potentially inconsistent definition of "facility". Omara appears to only have measured well pad sites and often refers to them as facilities or "facility-level", eg p.2102 starting just before Figure 1 "Among the routinely producing well pad sites, absolute facility-level CH4 emission rates varied by more than 3 orders of magnitude..." while the current manuscript appears to differentiate between well pad sites and facilities, where the latter have the potential to emit plumes at heights significantly above the assumed 1m AGL.

Can the authors please comment on the origin of the 2.2 g/s value in the Omara paper, as well as clarify the dif-
ferentiation between "wells" and "facilities" in their manuscript versus the Omara paper.

Comment 2: Can the authors please comment on their use of a constant emission rate of 0.59 g/s for all well pad sites in light of the text in Omara et al (2016, quoted above) stating that "...absolute facility-level CH4 emission rates varied by more than 3 orders of magnitude, with UNG sites exhibiting generally higher CH4 emissions (range: $0.85 \pm 0.40$ ($1\sigma$) to $92.9 \pm 47.5$ ($1\sigma$) kg/h) ..."

Thank you!

---

## Short Comment (SC2) · 2 Jun 2017

**Comments from the BC Oil and Gas Commission**

**on**

**Mobile measurement of methane emissions from natural gas developments in Northeastern British Columbia, Canada**

**Jun. 1, 2017**

The British Columbia Oil and Gas Commission (Commission) is the provincial regulator for the oil and gas industry. Depending on the activity the Commission is either the primary regulator, or works with other regulatory agencies to ensure activities are managed for the benefit of British Columbians. In August 2016, the province released the BC Climate Leadership Plan (CLP) which set a goal to reduce methane emissions from the upstream natural gas sector by 45 per cent below 2014 levels by 2025 from extraction and processing infrastructure built before Jan. 1, 2015.  The Commission is working with the B.C. Government to determine how to effectively meet this CLP goal.

The Atmospheric Chemistry and Physics discussion paper is of considerable interest to the Commission. Therefore, we have reviewed this discussion paper to determine if the findings agree with the regulator's extensive understanding of the oil and gas sector from the perspectives of protecting public safety, respecting those affected by oil and gas activities, conserving the environment, and supporting resource development.

Relevant to this discussion paper is that the Commission performs 4,000 to 5,000 inspections per year on oil and gas infrastructure and if methane releases are identified during an inspection, deficiencies are noted and industry is required to take corrective action. Also, routine checks on wells for surface casing vent flow are performed and if significant leaks are found industry is required to take corrective action.

In reviewing this discussion paper, considerable discrepancies were noted between the study findings and the Commission's understanding of oil and gas infrastructure within B.C.  Our findings are as follows:

**Overall:**

- L**ocation of infrastructure:** The facility data downloaded from the BC Oil and Gas Commission has NTS or DLS coordinates which are accurate to approximately 400 by 400 area. The discussion paper should provide clarity on whether the NTS or DLS locations were used or if and how the study refined the locations.
- **Emissions attribution:** There are numerous situations where multiple permits are issued by the Commission at the same general physical location. The discussion paper does not address how this was handled. When a methane plume is detected the discussion paper should indicate how this is attributed to a source when multiple wells and facilities are attributed to the same geographic location. How was a single release anomaly tied to estimating releases that could be tied to multiple permits at the same physical location?

- **Emissions rates may be overstated due to the use of averages:** In calculating emissions, the STFX/DSF study assumed, even for facilities that had emissions detected just over 50 per cent of the time, that their leak rate was constant and ongoing. The study noted that, especially with venting emissions, the release of methane may not be constant. This assumption has high potential to lead to an overstatement of methane emissions.

Specific discrepancies within the text are as follows:

Page 8 line 22 Well status of:

- "Cancelled" means the well permit expired without drilling commencing. So these wells do not physically exist in the field and can not be attributed to the release of methane.
- "Well Authorization Granted" (WAG) means that a well has been approved, but drilling has not commenced. Therefore these can not be attributed to methane releases.

Page 8 line 23

It is difficult to understand how the text "for the class defined in the databases as Well Authorization Granted, most of which were somewhere in the stages of development during our visits" could be correct. While some wells with a status of WAG would have commenced drilling between the time the well data was acquired in July 2015 and the study completed Sept. 5, 2015, this number is quite small compared to the total number of wells with a status of WAG. While it is unclear when in July 2015 the researchers obtained well data from the Commission, if we assume the data was obtained on July 1, 2015, there were 1,797 wells with a status of WAG. Between July 1, 2015 and Sept.r 5, 2015, 146 of these wells commenced drilling. As this data is for all of northeast B.C., a subset of these wells are located in the study area. In any event, a maximum of 8 per cent of WAG wells were somewhere in the stages of development during the field visits and the remaining 92 per cent did not physically exist at the time of the study and therefore were incapable of emitting methane.

In conclusion, for page 8 line 22 the text should be revised from "25% for Cancelled" should indicate no releases from cancelled and "27% for well authorization granted" should read close to zero for well authorization granted.

Page 9, line 5

The text refers to a category of "Undefined". It should be noted the term "Undefined" is not used to describe the well status (Well Authorization Granted, Drilling, Cased, Completed, Active, Cancelled, Suspended, Abandoned). "Undefined" is used to describe the well operational status (Production, Injection, Disposal, and Observation). For example, a cased well would have an operational status of undefined since it was never completed. In addition, undefined is used for the well fluid type (Gas, Oil, Multiple Gas, Multiple Oil, Multiple Oil and Gas or Water) if a well has not flowed in order to define the fluid type. For example, a well that was completed, but did not flow when tested would have an undefined fluid type. An active water disposal well would have a status of ACTIVE WATER DISPOSAL, not UNDEFINED.

Page 11 Line 11 to 18

The development of the MDL or release rate in the study involves significant uncertainty which is not adequately discussed in the text. Further information should be provided on the laboratory experiments used to determine a mean level of dilution of 70 per cent to demonstrate "realistic field conditions" and should include the range of results from those experiments.

Page 11, line 19 to 32

NOAA states that the Gaussian dispersion model is recommended as a teaching tool to understand basic concepts and does not recommend its use for dispersion studies. This paper should answer the question as to why this particular model was used when there are a multitude of other dispersion models to choose from.

Regardless of the dispersion model used, a sensitivity analysis should be completed for the main inputs used for the analysis in this study. As currently written, it is unclear which meteorological inputs (wind speed, wind direction, temperature, etc.) the researchers used, and whether they were representative of the region. Dispersion modelling can be highly sensitive to input parameters, and as such a further discussion of this uncertainty should be included, especially as the outputs from this modelling are used to determine as the release rate and to estimate a regional emissions inventory.

In conclusion, for Page 11 (lines 11 to 32), the technique used to develop the emission factor of 0.59 g/s is questionable.

Page 12, line 20

The term "facility" in the Omara study refers to the sum of wells and equipment at a multi-well site. Facility type as outlined in Figure 8 of this study is not the same as defined in the Omara study. There is no basis for using the emission factor 2.2 g/s in this discussion paper.

**Conclusion and Recommendation**

The fact significant quantities of emissions were attributed to wells that do not exist (i.e. 25 per cent of cancelled wells were reportedly emitting) calls into question the accuracy and validity of the discussion paper. Also, the basis for determining emission factors used in this discussion paper is highly questionable - therefore, this study should not infer that the estimates constitute an emission inventory that could be compared with what is reported under the Greenhouse Gas Emission Reporting Regulation. The Commission would welcome further dialogue to improve this study prior to publication.

---

## Author Comment (AC1) · 15 Jul 2017

Thank you to the reviewers for their positive and valuable comments. We are particularly grateful for the compliments about the quality of preparation, organization, and writing that went into this study. Since the submission of this manuscript, there has been an independent regulator-sponsored study for the same hydrocarbon resource (Montney) at an upstream development just across the provincial border in Alberta. This study strongly validates the CH4 emission patterns we saw in our work. Not only were the emission frequencies almost identical, but also the volume estimates were very much inline with ours. We are excited to incorporate details of that study into our manuscript to both strengthen and validate our methods and results.

We have addressed each referee and short comment individually below. Revised figures are included here, and we have shown all significant changes to the manuscript text (in colour). We believe that these changes have resulted in an improved manuscript.

**Response to Anonymous Referee #1 – RC3**

We very much appreciate the reviewer's comments, and are encouraged by the positive feedback and recommendation for publication. Please see below for our response to this review.

*General Comments*

*In "Mobile measurement of methane emissions from natural gas developments in Northeastern British Columbia, Canada", Atherton et al. describe with lucidity and apply with care an improved mobile survey technique for identifying methane leaks in an understudied region of Canada's oil and gas fields. The measurements are used to probe which aspects of the oil and gas infrastructure in the portion of the Montney region surveyed are most likely to emit methane. A conservative estimate of the bottom up inventory for entire Montney development is calculated and compared against state-based estimates, which is the most uncertain part of the analysis. The manuscript clearly describes the measurement and analysis techniques, highlights the limitations of the approach, and contextualizes the results nicely. I recommend this manuscript for publication in Atmospheric Chemistry and Physics with only minor changes.*

Thank you to the reviewer for this overview of our manuscript. We have made all minor changes to the manuscript that are addressed in the Specific Comments section below.

*Specific Comments*

*Line - Comment*

*p.2, 1.13 - "ostensibly less environmental impact" – People have been more concerned*

*about water-based impacts of hydraulic fracturing than those of coal, so restating this perceived advantage to be specific to atmospheric drivers of climate might be more accurate.*

We agree, and have changed the wording of this in the manuscript to be more specific about the environmental benefits related to atmospheric greenhouse gas emissions.

"For this reason, natural gas has been deemed a transition fuel on the path to renewable energy because it allows for continued fossil fuel exploitation while emitting a seemingly smaller amount of greenhouse gases."

*p.3, 1.13 - "super-emitters, and reduction" should be "super-emitters and reduction"*
We agree and have made this change in the manuscript.

*p.3, 1.26 - "significantly, with thousands" should be "significantly with thousands"*
We agree and have made this change in the manuscript.

*p.4, 1.3 - "August 14 2015 and September 05 2015 we" should be "August 14, 2015, and September 5, 2015, we"*
We agree and have made this change in the manuscript.

*p.8, 1.20 - "probably" should be "probable"*
We agree and have made this change in the manuscript.

*p.9, 1.7-8 - Indeed, accurate infrastructure inventories can be difficult to maintain. This statement seems to indicate that the correlations were not what was expected, which led to suspicion of the infrastructure inventories. Could you rephrase this statement to describe the limitations on analysis that uncertainties in the inventory induce?*
We have removed some lines from this part of the discussion, and we have instead added some text to the Methods section (under *2.3 Emission Source Attribution*) clarifying uncertainties in the acquired infrastructure inventory.

"When possible, we attempted to validate the infrastructure locations in the database during our surveys. The locations of the majority of well pads and processing facilities appeared to be accurate, however the statuses may not have been up to date. For example, well pads recorded as "abandoned" in the database, occasionally still had infrastructure present. Although we could not verify the locations of all infrastructural sources from public roads, we concluded that in most cases, the infrastructure database locations appear to be correct, but the operational statuses might not have been up to date."

*p.7, 1.25 - "FLIR" is first used here, but the acronym is first defined on page 10. Could you please reorder?*
This change has been made to the manuscript.

*p.12, 1.20-1 - "Montney based" should be "Montney-based"*
We agree and have made this change in the manuscript.

*Figure 2  - If I understood correctly, industrial sources were passed on multiple routes. Could these dots and bars be color-coded (with colors from Figure 1) by the route on which the source was observed?*

For clarity we have re-created this graph to show detection distances on each route. Below is the revised graph and caption.

[Figure]

"Figure 2: Mean distance from infrastructure while surveying each of the six routes listed in Figure 1. One standard deviation from the mean shows the range of distances at which we were sampling downwind of infrastructure."

*Figures5, 6, 7 – Please add to the caption the meaning of the grey-shaded areas around the line.*

In response to a comment from Anonymous Referee #2 (below) we have illustrated these data using bar graphs instead.

**Reply to Anonymous Referee #2 – RC1**

We thank the reviewer for their constructive review of the manuscript. We have made all of the specific recommended changes. Please see below for our response to each of the comments.

*General Comments*

*\* The manuscript is extremely well written. \* This paper addresses an important need in the community with a practical and well-described method for estimating emissions rapidly and on a broad scale. \* While I understand that there was not an opportunity to benchmark the estimates against other methods of emissions estimation, the lack of validation remains a significant weakness. I nevertheless recommend publication, but this caveat should be recognized at key steps in the analysis. \* The largest omission from*

*the paper is the lack of any uncertainty estimate for the emissions from the region. Some effort should be made to rectify this in the final paper. \* I don't understand the use of linear regressions (with variable slope and offset) for the detection rate estimates. Justification of why this analysis should be used over the simple calculation of rate = emitting sources / total sources should be provided, or the authors should revert to the simpler analysis.*

We appreciate the reviewer's general comments. The reviewer's concerns surrounding both uncertainty estimates and the linear regression plots are dealt with more explicitly in the Specific Comments section. We have addressed these comments in detail below.

***Specific Comments***

*- P1 L17: emissions estimates for the Montney development does not have an uncertainty estimate. It is difficult to interpret the emission results without an uncertainty associated with it.*

In our study we have made a minimum emissions estimate by combining the minimum detection limit of our applied method with our calculated emission frequencies for the infrastructure in the survey area. We expect that the total CH4 emission volume for the area is higher than our reported estimate.

A regulator-sponsored FLIR study was released at the same time we submitted our manuscript to ACP (GreenPath (2017)). The study was independent of ours, but took place in the Alberta portion of the Montney formation (the same play that is being developed in the field area of our study). The study by GreenPath Energy reported almost identical emission frequencies and emission volumes as we calculated for our field area. The results of our study reinforce the emission patterns of the GreenPath study across a larger sample size.

We have added the following text to section *3.4 Methane Emission Inventory Estimate* of our manuscript to address how this newly released study validates our method of volume estimation.

"Our emission frequency calculation for Active wells (0.47) was very similar to the emission frequency of 0.53 that was recently calculated in the Alberta Montney near Grande Prairie (GreenPath, 2017). Our method of calculating emission frequencies is corroborated by this recent FLIR study in the Alberta Montney, which increased our confidence in using emission frequency calculations to estimate a minimum CH4 inventory for the development."

*- P5 L1 - 10: The authors state that they are using excursions in the eCO2:eCH4 ratio (<150) as indications of natural gas emissions. However, I would imagine that other sources of CO2 could add noise to this ratio (especially since there are other vehicles that contribute to excess CO2). Figure 3 further indicates this issue. A fairly obvious alternative would be to use the same RMRI algorithm and use eCH4 > threshold as a criterion for when emissions are detected. It would be helpful if the authors could*

*provide some more justification why the ratio eCO2:eCH4 is a better metric than simply eCH4.*

The method of using excess ratios (particularly eCO2:eCH4) for plume source attribution in an upstream oil and gas environment is described in Hurry et al. (2016). We have added the following text to the manuscript in section *2.2 Identification of Natural Gas Emissions* to clarify that a detailed explanation of the method can be found in this paper.

"This eCO$_2$:eCH$_4$ approach has proven to be a useful fingerprinting tool in oil and gas environments because a single ratio value can help elucidate the presence of multiple emission source types. In this study, we follow a procedure similar to Hurry et al. (2016), and a detailed explanation of the method is described in that paper."

*- P5 L10-12: "Our optimal RMRI was taken to be the point at which anomalies were maximized, but also where we avoided the rapid noise-associated increase associated with extremely short RMRIs": in practice, how was this optimization performed? It appears to be a subjective choice. Is this true? It would be preferable if the choice was made objectively using quantitative criteria; it would also be preferable to have the same algorithm be used for all surveys.*

We did not choose the RMRI value for each survey subjectively. The optimization was performed with an algorithm that was applied to all surveys individually. We have added the following figure and associated text to the paper to clarify the quantitative process we used to determine the RMRI for each survey. Please see the figure, caption, and revised text below.

[Figure]

"Figure 2: Example of a regression plot that demonstrates the optimization process we used to calculate an RMRI for each survey. The RMRI for each survey was chosen where the two linear regression lines intersect."

*- P5 L 18-19: "Combustion values were also recorded along the routes when eCO2:eCH4 exceeded 1000, and were related to vehicle tail-pipe emissions and industry". What does 'combustion values' mean?*

This sentence has been re-worded in the manuscript to better explain how we filtered out emissions related to combustion.

"We also detected occurrences of combustion emissions along our survey routes, and we differentiated them from other emission sources by filtering out all values where eCO2:eCH4 > 1000. Combustion-related emission sources include vehicle tailpipe emissions and industry (ex. power generation)."

*- P5 L24-25: "because ratios are more conservative than concentrations in valleys and other areas where pooling of gases is common, and fewer false positives are likely" - doesn't the RMRI algorithm take care of slowly varying concentrations of CH4? It would be good to demonstrate clearly why eCO2:eCH4 is an advantage; if one were to reproduce this method at a larger scale, it would be good to provide clear understanding of why the CO2 concentration is required.*

It is possible that eCH4 would have been sufficient and may well have given similar results with few false positives. However, the excess ratio technique is established to be more useful in areas of complex upstream geochemistry to partition a number of emission source types (please see answer to comment *P5 L1-10* for explanation and reference to Hurry et al. (2016)). We did not resolve multiple peaks within the excess ratio density plots (Fig. 4 in the revised manuscript), which we would expect to see if there were multiple source types throughout our surveys. The excess ratio technique provided confidence that the source types are related to the infrastructure to which we were proximal during our surveys.

*- P5 L28-30: why was the value 150 selected? What is the effect of this selection on, for example, the emissions estimate, the number of emitters detected, the detection limit, etc. Similarly, what is the effective limit on detection of the system, in units of eCO2:eCH4?*

The value of 150 was selected based on peaks in the eCO2:eCH4 density distributions (Fig. 3). Although there is not a clear peak on each graph, many of the routes showed leveling out of the "natural" peak (~215) near 150-175. We chose 150 to be conservative, and it acts similarly to setting a methane excess threshold. Since our survey routes were focused in areas of dense oil and gas development, the elevated density of emissions with eCO2:eCH4 values <150 were interpreted to be from oil and gas related sources. The value of 150 was also considered to be conservative enough to exclude diluted CH4 from natural sources. Also, the exact ratio threshold often does not affect the number of plumes detected, but rather the width of the plume (duration while surveying), which is not pertinent to this study.

*- P6 L7: are there any estimates of cattle emission in this region that could be included?*

We were unable to retrieve this information for the fieldwork area and dates. However, our use of a 50% emission persistence threshold for identifying emitters likely rules out the possibility that we included emissions from livestock in our calculations.

*- P7 L10: how is this probability defined? Per mile? Per second? For the whole route? This isn't clear.*

This probability was defined for the whole route. We have now clarified in the manuscript that we calculated the probability of false plume detection for the entire Control Route.

*- P7 L1-5: The kernel density plots do not have a clear knee below 215. Where is 150 on this plot? why was 150 selelected, and not 125 or 175, for example?*

Please see answer to comment *P5 L28-30.*

*P7 L16-20 and Fig 4. Was wind direction used to evaluate whether a plume should have been detected from the green well pads? Are the databases of well locations up to date? Was there an effort to corroborate locations with on-ground survey or satellite imagery?*

The source location databases were up to date at the time we retrieved them (July, 2015). Locations of the majority of sources in the database near our surveys were verified during the on-ground survey campaigns. A section has been added to the manuscript about the uncertainty in infrastructure inventory in response to a comment from Anonymous Referee #1 *p9 1.7-8*. We have also reworded the caption of Figure 4 (now Fig. 5 in revised manuscript) for clarity.

"Figure 5: A subset of infrastructure locations that we surveyed during our field campaign in attributed form. This figure serves as an example of how we attributed wells and processing facilities to on-road plumes. Grey lines represent the survey route. In this case 31 wells or facilities were surveyed, and we used our attribution technique, which accounts for wind direction and distance to source, to determine whether or not these wells and processing facilities were probable emission sources.

*P7 L32: "it had to have > 50% emission persistence." Similarly, did persistence include wind direction? In other words, did persistence include whether the potential source was upwind of the vehicle at the moments the vehicle passed by?*

Yes, our calculation of emission persistence included only the sources we had sampled. And in order for a source to be considered sampled, at least three successive datapoints had to be downwind and within 500 m of the infrastructure in question. We have clarified this in the following section of the manuscript:

"In this study, emission persistence is defined as the number of surveys on which a CH4-enriched plume was attributed to a piece of infrastructure, divided by the number of times we surveyed that infrastructure in the downwind direction. A plume was only attributed to a piece of infrastructure if we recorded three or more successive CH4-enriched measurements within 500 m in the downwind direction of the source. And in order for a piece of infrastructure to be classified as an emission source, it had to have > 50% emission persistence."

*P11 L8: "concentrations will decrease exponentially away from a release source": the dependence on distance is not exponential. Gaussian plume models predict something*

*like ~1/d to 1/d^2, for example.*

Thank you for pointing this out. We have removed "exponentially" from this sentence in the revised manuscript.

*P11 L11-18: Wouldn't nearby plumes (with faster time signatures) be diluted more than more distant plumes? And wouldn't the peak area (in time) be conserved for short pulses? This is a very big adjustment of the concentrations and therefore the emissions. Did you use peak height or peak area to estimate emissions?*

Gaussian plume analysis depends on plume centerline concentrations, not widths.

*P12 L9: Rather than using the MDL as the average estimate of emissions, wouldn't it be possible to actually craft an estimate of emissions given the plume dispersion model and estimated distances?*

The process of calculating emission rates using Gaussian plume dispersion for each individual datapoint is computationally intensive because of the amount of measurements collected. The technique of applying volume estimates to mobile survey data was not developed at the time we processed these data. Our research group is currently developing a similar technique of volume estimation, but this will be part of a separate study and ground validation is still required.

*p12 L28: It is important to include some uncertainty estimates for the emissions estimate. Even a simple low and high estimate of error is better than nothing. For example, the estimates of errors on the slope of the active wells could be used to bound the estimate.*

Please see our answer to comment *P1 L17* from Anonymous Referee #2 for an explanation of added text about method validation. The linear regression plots have also been changed to bar graphs in response to comment on Fig. 5, 6, and 7.

*p14 L9: It's not clear how this method identifies super emitters, since the authors do not present a clear method for quantifying emissions and identifying the largest emitters. How does this method help identify the largest emitters?*

This section of the manuscript is referring to the benefits of using an on-ground detection method that surveys a large fraction of infrastructure throughout the development. In comparison to emission factor inventory estimates, we are more likely to have captured emissions from super-emitters. We have added the following text to section *3.1 Measured Gas Signatures* to address our results relative to what would be expected from super-emitting sites:

"We did not see any CH4-rich plumes that would be characteristic of a super-emitter. This is evident by the fact that the maximum raw CH4 value we recorded was low (8.148 ppm). These low emission magnitudes are inline with results from GreenPath Energy (2017), which used FLIR cameras to assess emission sources in the Alberta portion of the Montney formation."

*Fig 5: In some panels (e.g., the top panels), the regression lines do not pass through zero. This doesn't make any physical sense. Why should there be a threshold for number of wells surveyed below which no emissions should occur? Why would there be no emissions for surveys with fewer than 60 wells surveyed? I don't understand the rationale for a linear regression. Why not simply ratio the total number of sites with emissions / total number of sites surveyed across all surveys for each category? This would make more intuitive sense. Alternatively, the linear regressions could be forced through zero, which would be similar.*

*Fig 6 and 7: similar comments to above for Fig. 5.*

We agree and have changed the linear regression plots to bar graphs which show the percentage of infrastructure emitting for each source-type. Please see the graphs and captions below. We have also made minor changes to the manuscript text accordingly.

[Figure]

"Figure 6: Emission frequencies for each well mode type for all surveyed infrastructure on each route. These emission frequencies were considered in our total emissions inventory calculations."

[Figure]

"Figure 7: Emission frequencies for each well operation type for all surveyed infrastructure on each route. Certain operation types for which we did not have a

representative sample are not included (such as Injection, Disposal, and Observation wells)."

[Figure]

"Figure 8: Emission frequencies for each facility type for all surveyed infrastructure on each route. These emission frequencies were considered in our total emission inventory calculations."

*Fig 8: Is the occurrance structure due to the fact that some areas were surveyed only three times, which did not allow for a 50% persistence point, for example? This set of plots is a bit confusing.*
(This is now Figure 9 in the revised manuscript). In this figure, "Occurrence" (y-axis) refers to the number of pieces of infrastructure emitting at each level of persistence (x-axis). The y-axis has been re-named to "Unique Wells/Facilities (n)" for simplicity. Below is the edited caption.

"Figure 9: The cumulative number of unique wells/facilities versus emission persistence (%) across all 30 mobile surveys. Persistence refers to the repeated tagging of a piece of infrastructure as a possible emission source based on the method of plume attribution we applied in this study."

*Fig 9: what do negative mean eCH4 excursions mean (gray bars of lower panels)?*
(This is now Figure 10 in the revised manuscript). We have removed the grey error bars from this figure. Below is the edited caption.

"Figure 10: Effect of infrastructure age and operator size on detected emissions. The size of the dots represents the number of samples taken. Red dots are those recorded at the 100% persistence level, green dots are at 50% persistence."

*Fig 10: could you add in the survey paths on this plot for reference?*
(This is now Figure 11 in the revised manuscript). We have chosen not to add the

survey routes because the size of the dots already represents the sample size in each area.

*Typographical error and other small comments*

*P1 L13-15: "older infrastructure tended to emit more often (per unit) with comparable severity in terms of measured excess concentrations on-road." - unclear; per unit? what is a unit? reword for clarity, please.*

"Unit" was referring to each individual piece of infrastructure. This has been reworded in the manuscript for clarity.

"Multiple sites that pre-date the recent unconventional Montney development were found to be emitting, and we observed that the majority of these older wells were associated with emissions on all survey repeats."

**Reply to Anonymous Referee #3**

We thank the reviewer for their thorough and detailed review of the manuscript text. Highlighted below are changes to the text we have made to address the suggested revisions.

*The authors present data and analysis from six mobile measurement surveys in the Montney formation which include methane emission concentration and rate information from 1600 passes near wells. The routes were surveyed 3-6 times each and designated as new wells, old wells, and a control. The authors use the methane and CO2 concentration and meteorology data to calculate emission rates of methane from wells. They analyze the data using online well number, production, age, etc. information to show which types of wells or activities emit most or most often. And finally, they compare their results to available data from recent studies in other formations in U.S. Collection of mobile data, especially when one is at the whim of wind to assure downwind of well measurements, is no easy task. The authors have conducted a great survey of sites in the Montney formation. This study is exactly the type of research that is needed to clarify and quantify the emission rates of methane from different formations and sources. The authors have done a lot of work and the publication of this paper (especially with the availability of the data upon request, as noted at the end of the manuscript) will be a great addition to the current body of knowledge on methane emissions from oil and gas sources. However, there is some more analysis, organization, and sentence structure improvement that is needed for this paper before publication. Please see my General and Specific comments below:*

*General Comments*

*1. Various groups have used different approaches to quantifying methane emission rates (e.g., EPA's OTM 33 method, use of different tracers for close or far quantifications using*

*the Tracer Ratio Method, reverse plume modeling, etc.). One of the things that all the methods above have in common is method validation. It seems that the authors of this paper have not conducted any method validation studies. This is a major weakness in the study. I would recommend that a quick methane and CO2 release study and measurement be added to the paper. However, I understand that time and funding may not be available to do this. Instead, I suggest the authors do a detailed uncertainty analysis (maybe even add a section to the paper) where they discuss and calculate a theoretical uncertainty for their measurements and calculations. The authors have a short section on this, but since no method validation has been done, the uncertainty analysis should more exhaustive.*

Please see our response to comment *P1 L17* from Anonymous Referee #2 for information on method validation and how our calculations are very similar to results from a recent study at a nearby oil and gas development accessing the same hydrocarbon formation (GreenPath, 2017).

The primary objective of our study was to collect data on emission frequencies and to establish what infrastructure types emitted most frequently. Minimum volumetric estimates were included, but were not the main focus. Calculating emission frequencies for every oil and gas development is important because it determines the number of wells/facilities by which emission factors should be multiplied in order to achieve an accurate emissions inventory estimate.

We have added the following text to section *1 Introduction* of our manuscript to clarify that emission frequency calculations were the main objective of this study.

"In this study we used a multi-gas ($CO_2$, $CH_4$) mobile surveying method that uses ratio-based gas concentration techniques and wind data to detect and attribute on-road $CH_4$-rich plumes to the infrastructural sources of natural gas developments in northeastern British Columbia, Canada. Our primary interest in this study was to determine the frequency of emissions, and the relationship between emissions and specific classes of infrastructure."

*2. Another point that is not clarified in this paper is the difference between measurements made from unconventional vs conventional wells. The authors make a distinction between new and old wells. The attribute the increase in the oil and gas activity in the area to the use of unconventional extraction methods. However, when they discuss the wells measured, they do not show any information on the unconventional vs conventional wells. Are all the wells measured unconventional?*
The area we surveyed in Northeastern British Columbia mainly produces unconventional natural gas. A large majority of gas wells we surveyed use unconventional techniques of extraction (hydraulic fracturing and/or horizontal drilling). We included one survey route that targeted an area of conventional oil development for comparison (Route 1). The increase in development in the area over the last decade has been from unconventional natural gas infrastructure (discussed in section *1 Introduction*). Information about what type of

infrastructure is on each route is included in section *2.1 Field Measurements*. And the difference in emission frequencies from oil and gas infrastructure is shown in Figure 8 (now Figure 9 in the revised manuscript) in the chart titled Well Fluid Type.

*3. The authors do not distinguish between short term operations and permeant emission sources in their calculations. This may be difficult to do, but at least a discussion of how these would affect the regional emission calculations should be added.*

In this study we look at emission persistence in terms of survey repeats. To be conservative in our method of identifying emitting infrastructure, we only tagged infrastructure as emitting if we detected CH4-enriched plumes within 500 m downwind at least 50% of the times we surveyed it. For many of the pieces of infrastructure we surveyed this means it was associated with a plume downwind on three out of six surveys. We have added text to clarify this in section *3.4 Methane Emission Inventory Estimates*.

"This value is likely a conservative estimate because it is the smallest value detected at our mean detection distance (319 m), and the majority of our emission detections occurred around this value (Fig. 3). It is also conservative because our method of attribution only considers the wells and facilities that were persistently associated with downwind plumes."

*4. Some of the writing in the paper is confusing. The sentence structures do not flow well. I have given some specific examples of this in the "Specific Comments" section, but strongly suggest the co-authors who were not directly involved in the writing of the manuscript read the paper and comment on sections. Sometimes it is easy for the authors to unintentionally disregard clarity as they themselves are so familiar with the subject of the study.*

We have combed the manuscript with this comment in mind and improved the phrasing as recommended in the "Specific Comments" section of this review.

*5. The authors use two different tenses and two different voices (active and passive) throughout the paper. I suggest choosing only one. Two different voices and tenses make it confusing for the reader and require re-reading of sections.*

We have made all necessary changes to move from passive to active voice.

*Specific Comments*

*1. Abstract: The writing style of the abstract does not lend itself to clarity. The flow of the sentences is not coherent. I suggest re-writing it for better clarity and flow. For example: "We also observed emissions from facilities of various types that were highly repeatable." is one of the sentences that is unclear and confusing. Or "This value exceed reported bottom-up estimates of 78,000 tonnes for all oil and gas sector sources in British Columbia, of which the Montney represents about 55% of production". The abstract starts very abruptly. I suggest rewording the first sentence.*

The following sections of the abstract have been revised for clarity, as well as

sections addressed in response to comment from Anonymous Referee #2 *p1 L13-15.*

"In August to September, 2015 we completed almost 8,000 km of vehicle-based survey campaigns on public roads dissecting oil and gas infrastructure such as well pads and processing facilities."

"Emissions from gas processing facilities were also highly repeatable."

"This estimate for the Montney area exceeds reported bottom-up estimates of 78,000 tonnes methane for all oil and gas sector sources in the province. Current bottom-up methods of methane emission estimates do not normally calculate the fraction of emitting infrastructure through thorough on-ground measurements. However, this study demonstrates that mobile surveys could be used to gather a more accurate representation of the number of emission sources in an oil and gas development. This study presents the first mobile collection of methane emissions from oil and gas infrastructure in British Columbia, and these results can be used to inform policy development in an era of methane emission reduction efforts."

*2. Page 1, Line 2: What do the authors mean by "incidence"?*
"Incidence" was used interchangeably with "emission frequency". This sentence has been reworded for clarity, and "incidence" has been changed to "emission frequency" throughout the text of the manuscript.

"This study examined the occurrence of methane plumes in an area of unconventional natural gas development in northwestern Canada."

*3. Page 1, Line 4: Are authors including all oil and gas locations in "development". I suggest clarifying this or using a different word.*
"Development" refers to areas where oil and/or gas is being extracted, and oil and gas infrastructure is dense. It has been changed in the abstract, and defined when it is first used in the manuscript.

"North American leaders recently committed to reducing methane emissions from the oil and gas sector, but information on current emissions from areas of unconventional natural gas extraction in Canada are lacking."

*4. Page 1, Line 6: The use of "infrastructural" here has the same problem as the previous comment.*
"Infrastructural" refers to oil or natural gas infrastructure, including wells and processing facilities. This has also been reworded in the abstract and defined in the manuscript for clarity.

"To attribute on-road plumes to oil and gas related sources we used gas signatures of residual excess concentrations (anomalies above background) less than 500 m downwind from potential oil and gas emission sources."

*5. Page 2, Line 5: What do the authors mean by "a petroleum system".*
A petroleum system is a term defining all the necessary geological components and processes required for the formation and accumulation of hydrocarbons.

*6. Page 2, Line 14: Please rewrite "Over a 100-year... (ICPP, 2014)" for clarity.*
This sentence has been revised for clarity.

"The radiative forcing of CH4 is greater than 30 times that of CO2 over a 100-year timespan."

*7. Page 2, Line 33: I have noted this in the abstract too. Please describe what you call "infrastructure".*
Please see answer to comment *4. Page 1, Line 6* above.

*8. Page 3, Line 1,2: Please re-write sentence for correct grammar.*
This sentence has been revised for clarity.

"Furthermore, it is important to note that emission frequencies may vary between developments because of operator best practice, or due to the properties of the geological formation that the hydrocarbons are being extracted from."

*9. Page 3, Line 13: Please define "super-emitters" and use appropriate references.*
This sentence has been changed to include all emission sources.

*10. Page 3, Line 26: Do the authors have some estimate of numbers of wells?*
We have revised this line discussing the increase in natural gas production to the following:

"These unconventional methods yielded 4-5 times more natural gas from the Montney formation than conventional techniques that were attempted prior to 2005. Since then, production of BC unconventional natural gas has increased significantly, with the Montney play being the largest contributor in the province (BC Oil and Gas Commission, 2012)."

*11. Page 4: The authors use the words unconventional and hydraulically fractured interchangeably. These two do not mean the same thing. Unconventional oil and natural gas extraction refers to both hydraulic fracturing and horizontal drilling.*
The use of "hydraulically fractured wells" has been changed to "unconventional wells" where appropriate throughout the text of the manuscript.

*12. Page 4, Line 8: Is 1Hz frequency the rate of data collection?*
Yes, it is the rate of data collection. This sentence has been reworded for clarity.

"In total we surveyed 7,965 km of public roads, with an average route length of 248 km. We collected gas concentrations and wind data at 1 Hz frequency while surveying."

*13. Page 4: What were the average distances from wells? If this data is available, can it be used with meteorology data for plume dispersion modeling?*
We calculated the average distance from wells and used this value with plume dispersion modeling to calculate our minimum detection limit in section *3 Results and Discussion* of the manuscript.

*14. Page 4, Line 14: Please re-write for correct grammar.*
This sentence has been reworded to the following:

"We surveyed four of the routes six times throughout the field campaign, and the two remaining routes (including the Control Route) three times each. We repeated surveys on multiple days to account for varying wind directions. Repetitions of each survey route included both morning and afternoon drives to incorporate varying atmospheric conditions. We also used the repeated survey data to obtain statistics on emission persistence."

*15. Page 4: Please note which routes the numbers are based on in Figure 1.*
We have referenced the route names from Figure 1 in this section.

*16. Page 4, Line 19: What do the authors mean by "raw" ?*
We used the term "raw" in this section to make clear that no processing was done to the atmospheric gas concentrations at this phase of data collection.

*17. Page 4, Line 23: What are wind speed units?*
The wind speed was measured in km/h. We have added this information to the manuscript.

*18. Page 4, Line 25: Since the authors have given the manufacturer of the other instruments used, why not indicate what type of GPS was used?*
The type of GPS used has been included in the manuscript.

*19. Page 4, Line 32: Please re-write "However, our surveys. . . unusable." for clarity.*
This sentence has been rewritten to the following:

"The survey routes in our study were multiple hours long each and were often routed through various land use types. For this reason, we did not use the traditional methods of calculating background atmospheric gas concentrations."

*20. Page 5: Were the same approaches used for both CO2 and CH4 data handling and analysis? Please add a few sentences to clarify this.*
Yes, we used the same method of data processing for all gas measurements collected (CO2 and CH4). We have added text to clarify this in the manuscript.

*21. Page 5, Line 10-12: Please add some statistical data.*
Please see answer to Anonymous Referee #2 *P5 L10-12.* We have added an example plot to explain our method of choosing the RMRI.

*22. Page 5, Line 20-21: Please re-write for clarity.*
We have rewritten this sentence to the following:

"We identified CH4 plumes from oil and gas infrastructure in areas where there were multiple successive datapoints with depressed eCO2:eCH4 values."

*23. Page 5, Line 21: What do the authors mean by "normal air"?*
The term "normal air" has been changed to "ambient air" in the manuscript.

*24. Page 5: What are some sources of CO2 in the area? As this can be a major concern in your calculations, please add a few sentences to address this.*
As detailed in Hurry et al. (2016), the ratio technique helps identify (and remove) measurements that are enriched with respect to CO2. We have included the following text in section *2.2 Identification of Natural Gas Emissions* to describe possible sources of CO2 emissions in the area:

"Variation of CO2 within the survey area was likely primarily a function of oilfield processes (emissions, engines, flares) because there was little industrial activity on the survey routes that was not related to oil and gas development."

*25. Page 6, Lines 1-2: Please re-write for clarity.*
We have rewritten this sentence to clarify.

"Otherwise, all in-place oil and gas infrastructure were considered possible emission sources."

*26. Page 6, Line 2: What do the authors mean by "developmental"?*
The term "developmental" meant that the well was under development. This term has been removed and this sentence has been reworded to the following:

"The infrastructure database included the well and facility locations, as well as various attribute data such as infrastructure types, statuses, and spud dates (drilling dates)."

*27. Page 6: Are there any large dairy operations in the area?*
We did not encounter any large feeding operations while surveying. We only encountered smaller farms for which a database of locations could not be obtained.

*28. Page 6, Line 15: Please re-write sentence for clarity.*
We have rewritten this sentence for clarification.

"We collected atmospheric gas concentration data along 30 surveys of six different routes. The routes ranged in length from 200 - 550 km, and the oil and gas infrastructure located on these routes was managed by more than 50 different operators at the time of surveying."

*29. Page 6, Line 16: I thought the authors used one route as control. Did they actually make measurements from oil and gas structures on this route and include them in the analysis? If yes, then should the designation not be changed?*

The route we used as a control had significantly less infrastructure. This allowed us to visually compare sections of the surveys near infrastructure, and sections far away from infrastructure. We only used the Control route datapoints > 5 km from any infrastructure to calculate the fraction of false positives.

*30. Page 6, Line 19: Following up on the previous comment, please give numbers of the differences in the oil and gas densities.*

The amount of infrastructure on each route (sampled and emitting) is listed in Table 1.

*31. Page 6, Line 30: What was the speed of the car during these measurements? This is important as it can have an impact based on the width of the plumes.*

The vehicle speed was variable due to the speed limits on the public roads we were surveying. Plume width was not incorporated into any of our measurements, including our estimate of leakage rate. For this reason we have not included vehicle speed in the manuscript.

*32. Page 6, Lines 31-32: What is the difference between 314 and 319 meter designations? Also, should this not be in the methods section instead of the results section?*

We calculated average distances between the survey route and infrastructure for two scenarios: the first being datapoints when we were sampling infrastructure (314 m), and the second being when we were detecting emissions from infrastructure (319 m). We have reworded these lines in the manuscript to clarify this point. These values are not in the methods section because they were calculated from the collected data and the locations within the infrastructure database.

*33. Page 7, Line 1: What do the authors mean by " In each, we see a peak of signatures near ~215 which is representative of natural"?*

This sentence has been reworded in the manuscript to the following:

"In each density plot, there is a peak where eCO2:eCH4 = ~220, which is representative of the ratio between ambient CO2 and CH4."

*34. Page 7, Line 5: relative to what?*

For clarity, we have reworded this sentence to the following:

"The kernel density plots in Figure 1 show that, in all of the survey routes except the Control, we see a population of CH4-enriched anomalies (less than the natural ratio of 220), that are the result of localized plumes from natural gas development."

*35. Page 7, Line 25: What are the other methods?*

"Other" was a typographical error that has been revised.

*36. Page 7, Line 27: Should this be associated?*
We have removed the word "associate" from this sentence.

*37. Page 7, Line 30: Please define what you mean by ". . . a piece of infrastructure. . ."*
The use of the term "infrastructure" in this manuscript refers to oil and gas related infrastructure such as well pads and processing facilities. This is described earlier in the manuscript in response to comment *4. Page 1, Line 6.*

*38. Page 7, Line 32-34: Please re-write for clarity.*
We have reworded this in the manuscript to make this point more clear.

"Our technique of background subtraction is tuned to resolve small, localized plumes, but it should be noted that atmospheric conditions have a significant…"

*39. Page 7: I suggest adding clarifying sentences like, Well pads were the most common oil and gas structures encountered/sampled during our survey (%# of total sites).*
We have added the following lines to help refine this section of the manuscript.

"Well pads were the most common type of oil and gas infrastructure sampled during our surveys (58% of total infrastructural emission sources)."

"Emitting infrastructure includes wells and facilities where at least half the transits past the well were associated with a CH4 plume in the downwind direction (50% persistence)."

*40. Page 8, Lines 5-8: Please re-write for clarity.*
We have reworded this in the manuscript to the following:

"Many previous fugitive emission detection studies do not replicate surveys, but repeated emission detections help build both confidence in detection, as well as statistics about emission severity and persistence through time."

*41. Page 8, Lines 14-17: Please use a consistent theme for capitalization.*
We have made changes throughout the manuscript so that all well/facility status and types are capitalized.

*42. Page 8, Line 20: Please replace the term "probably" with one with a more scientific connotation or even some statistics.*
This was a typographical error. "Probably" was mean to be "probable", and we have made this change in the manuscript in response to comment from Anonymous Referee #1 *p.8, 1.20.*

*43. Page 8: Please explain, clearly, what each category of wells encompasses. For example, does authorization mean that permit was granted? Was work on the pad started? Was temporary drilling part of the study or as noted previously was it excluded?*

We have added the following text to the manuscript to clarify the definitions of these terms where possible. Please also see our reply to Tony Wakelin's comment below concerning certain well statuses.

"The infrastructure inventory we obtained from the provincial regulator identified several statuses of wells including Active, Abandoned, Cancelled, Completed, and Well Authorization Granted (WAG). It should be noted that Cancelled means that the permit for the well has been cancelled, usually before drilling has begun. Similarly, wells with the status of WAG may not have commenced drilling at the time we completed our surveys. However, based on discrepancies noted in the field about abandoned infrastructure, the accuracy of the status information in the inventory database could not always be relied upon. Furthermore, we assumed that test drilling and nearby infrastructure in these locations might serve as potential emission sources as well, so we chose to include wells with these status types in our analysis. A well with a Completed status means that the well drilling was complete, and it was being prepped for production."

*44. Page 9, Line 27: 60 out of how many?*
The total number of active wells we sampled is listed in Table 2. However, we have added the total (676) to this line in the manuscript.

*45. Page 10, Line 2: Please reword ". . . less emission prone. . ."*
We have reworded this line in the manuscript for clarity.

"Infrastructure type is a potential driver of emission patterns, which supports studies that have found large discrepancies in emission factors between valves used in different regions of the US (Allen et al., 2013)."

*46. Page 10, Line 20-32: This paragraph does not belong in this section. I suggest either deleting it or moving it to a more appropriate location.*
We have left the first line of this paragraph in this section of the manuscript. The rest of the paragraph has been integrated with the final paragraph in section *4 Conclusion*, and now reads as follows:

"Methane emission reduction strategies for large natural gas developments such as the Montney should focus on first locating super-emitting sites, and then follow up with site-specific emission techniques such as FLIR cameras. This strategy would support LDAR already in place, in a way that would minimize cost to individual operators. It would also focus the attention on the problematic infrastructure and operators, and does not share the cost burden across companies that have already invested heavily in emission reduction technology and leading best practice. It is feasible to detect super-emitters through exhaustive survey campaigns, even …"

*47. Page 12: Please give a more detailed (method definition, details, and statistics) of the setup of your calculations.*
Where possible, we have added further details to this section of the manuscript.

However, we feel that the calculations are made clear in Table 2. We did notice a typographical error in the Emission Volume column of Table 2, which has since been amended.

Table 2. Emission volume calculations for all surveyed infrastructure, and also extrapolated to account for all wells and facilities within the BC portion of the Montney formation. Our minimum detection limit (MDL) of 0.59 g/s was used as the emission factor for wells. Facility emission volumes are from Omara et al. (2016) because our sampling from facilities was probabilistic due to emission height variance.

| Type | Infrastructure n | Emission Freq (%) | Emission Volume (tonnes/year) | Emission Total (tonnes/year) |
|---|---|---|---|---|
| **Surveyed Wells** | | | | |
| Active | 676 | 47 | 18.6 | 5910 |
| Abandoned | 228 | 26 | 18.6 | 1103 |
| Cancelled | 130 | 35 | 18.6 | 846 |
| Completed | 64 | 30 | 18.6 | 357 |
| Surveyed Facilities | 265 | 32 | 70 | 5936 |
| Total CH$_4$ volume | | | | 14152 |
| | | | | |
| **Montney Wells** | | | | |
| Active | 5294 | 47 | 18.6 | 46,280 |
| Abandoned | 2149 | 26 | 18.6 | 10,392 |
| Cancelled | 1989 | 35 | 18.6 | 12,948 |
| Completed | 582 | 30 | 18.6 | 3248 |
| Montney Facilities | 1742 | 32 | 70 | 39021 |
| Total CH$_4$ volume | | | | 111,889 |

*48. Page 13, Line 5: Have the number of wells changed since 2012? Would this affect the calculations in this paper, especially when dealing with the comparison to other sites/studies?*
Yes, there was most likely a change in the number of active wells between 2012 and the time these surveys took place in 2015. Unfortunately, the most recent regional CH4 emission estimate we could find for the area was from 2012. We have added the following text to section *3.4 Methane Emission Inventory Estimate* of the manuscript to clarify this discrepancy and how it affects our comparison to the provincial estimate.

*"It should be noted that the most recent available CH4 emissions inventory from the province was from 2012, and that increased development and production from the Montney since then may have increased what the regulator would expect to see from this development. However, the 2012 estimate was the most recent applicable emissions estimate we could locate to compare our estimate to."*

*49. Page 13: Please add a discussion of possible reasons for the differences in this study and others noted here. Uncertainty range? Different basins? Different measurement approaches?*
We have added the following text to explain the differences in measurement

approaches:

"Although airborne measurement techniques are not ideal for locating exact emission sources, they are well-suited to calculate total emission volumes for entire regions so long as other emission sources (such as agriculture) can be accounted for, which they were in the studies listed above. The top-down nature of mobile surveying large amounts of infrastructure allows for a comparison between our CH4 volume estimate and those of Peischl (2016) and Karion (2015)."

*50. Page 13, Line 29: Please give numbers.*
We have included the emission frequencies here. This line has been revised to:

"Abandoned wells were also associated with emissions at 26% of the 228 sites we sampled, and we located a group of aging infrastructure (> 50 years old) that was emitting every time we sampled downwind."

*51. Please revise the Conclusion. It needs more specific numbers and information. Also, the addition of super-emitters at the end does not make sense as this paper was not directly making measurements from such sites based on the previously discussed statistics.*
To maintain the brevity of the paper we have decided to not include more specific results in the Conclusion. As discussed in response to comment *P14 L9* from Anonymous Referee #2, the mobile survey method is ideal for detecting super-emitters. However, our results were not indicative of the presence of super-emitting sites in the BC Montney, and our results mirror the results found in an independent study by GreenPath Energy (2017).

*52. Figure 1: Is it possible to add the location of the wells here as a light gray background? It would be helpful in visualizing the type of routes. Also, please make sure that your designations of routes in this figure and the paper are the same. After reading through, I found TABLE 1 in Tables. Do authors mention this table in the text of the manuscript?*
The map scale does not allow for the infrastructure locations to appear as individual points. The routes designations in Figure 1 are correct, and Table 1 is now referred to in the text of the manuscript in the following sections: *2.1 Field Measurements*, and *3 Results and Discussion*.

*53. Figure 2: What are 88 industry- defined areas?*
We have revised this figure to show the detection distances on each route. Please see response to Anonymous Referee #1 *Figure 2.*

*54. Figure 3: This is not a comment on this figure, but in looking at this and other figures, having a table with route numbers, names, and characteristics would be very helpful. Something like Table 1.*
This information is included in Table 1.

*55. Figure 4: Please revise caption to explain graph better. What are the gray lines?*
The gray lines are surveyed roads. This is now explained in the caption.

*56. Figure 5: Please re-write caption for clarity. Also, the addition of the uncertainty discussion as noted before, will help this figure.*
*57. Figure 7: Why are there zero-zero points in this graph? Although physically a zero-zero point makes sense, I do not think the addition of the points is statistically sound.*
We have re-plotted the regression plots as bar graphs. Please see response to comment from Anonymous Referee #2 *Fig.5,6,7.*

*58. Figure 9: Please add numbers in the increasing sample size legend. Were any of the wells in this area re-worked? This will change the definition of well age in this discussion.*
We have not added numbers to the increasing sample size legend because each graph in the figure has a slightly different scale. However, one legend for sample size without numbers is sufficient because it is only meant to show the relative number of times we sampled infrastructure in each category.
We did not have information on whether or not wells were re-worked.

**Reply to Brian Crosland – SC1**

We would like to thank Brian Crosland for his questions about the content of our study. We have addressed both of the comments below. No significant changes were made to the manuscript in response to this review.

*Comment 1: -Page 12 Line 20 refers to Omara et al and quotes a "natural gas facility emission volumes of 2.2 g/s". -Reading through the Omara paper it is not immediately clear where this value originates. As per the Omara abstract: "mean facility-level CH4 emission rate among UNG well pad sites in routine production (18.8 kg/h (95% confidence interval (CI) on the mean of 12.0-26.8 kg/h))" -Note that 18.8 kg / h works out to 5.2 g/s.*
*-Clouding the issue is a potentially inconsistent definition of "facility". Omara appears to only have measured well pad sites and often refers to them as facilities or "facility-level", eg p.2102 starting just before Figure 1 "Among the routinely producing well pad sites, absolute facility-level CH4 emission rates varied by more than 3 orders of magnitude..." while the current manuscript appears to differentiate between well pad sites and facilities, where the latter have the potential to emit plumes at heights significantly above the assumed 1m AGL. Can the authors please comment on the origin of the 2.2 g/s value in the Omara paper, as well as clarify the differentiation between "wells" and "facilities" in their manuscript versus the Omara paper.*
We will seek to verify the definition of "facility" with Omara and perhaps a corrigendum can be issued that clarifies. We have used the emission rates that we can best tell are accurate for a natural gas facility in our study area without further explanation. Furthermore, as the BC OGC have pointed out in their comment below, many well pads in the area we surveyed have multiple types of

infrastructure (wells and processing facilities) on the same well pad. It is therefore reasonable to assume that Omara's estimate of facility-level emissions is likely a realistic comparison to the locations classified as "facilities" in our study.

*Comment 2: Can the authors please comment on their use of a constant emission rate of 0.59 g/s for all well pad sites in light of the text in Omara et al (2016, quoted above) stating that "...absolute facility-level CH4 emission rates varied by more than 3 orders of magnitude, with UNG sites exhibiting generally higher CH4 emissions (range: 0.85 ± 0.40 (1σ) to 92.9 ± 47.5 (1σ) kg/h) ..." Thank you!*

Our CH4 volume calculation is an estimate of the *minimum* CH4 emissions in the area. As is outlined in our manuscript, we used emission frequencies of sources that we identified to be emitting persistently. To provide a conservative estimate of emissions, we applied our minimum detection limit to the fraction of persistent emission sources in the area. For this reason we have stated in our paper that our emissions inventory likely underestimates the real total CH4 emissions for this area.

**Reply to Tony Wakelin – SC2**

We would like to thank Tony Wakelin from the BC Oil and Gas Commission for his interest in our manuscript. It is helpful to have critical feedback from members of the provincial regulatory organization, as they often have important knowledge about the inner-workings of the local oil and gas industry. We have addressed each comment below, and have included the related edits made to the manuscript.

*The British Columbia Oil and Gas Commission (Commission) is the provincial regulator for the oil and gas industry. Depending on the activity the Commission is either the primary regulator, or works with other regulatory agencies to ensure activities are managed for the benefit of British Columbians. In August 2016, the province released the BC Climate Leadership Plan (CLP) which set a goal to reduce methane emissions from the upstream natural gas sector by 45 per cent below 2014 levels by 2025 from extraction and processing infrastructure built before Jan. 1, 2015. The Commission is working with the B.C. Government to determine how to effectively meet this CLP goal.*

*The Atmospheric Chemistry and Physics discussion paper is of considerable interest to the Commission. Therefore, we have reviewed this discussion paper to determine if the findings agree with the regulator's extensive understanding of the oil and gas sector from the perspectives of protecting public safety, respecting those affected by oil and gas activities, conserving the environment, and supporting resource development.*

*Relevant to this discussion paper is that the Commission performs 4,000 to 5,000 inspections per year on oil and gas infrastructure and if methane releases are identified*

*during an inspection, deficiencies are noted and industry is required to take corrective action. Also, routine checks on wells for surface casing vent flow are performed and if significant leaks are found industry is required to take corrective action.*

*In reviewing this discussion paper, considerable discrepancies were noted between the study findings and the Commission's understanding of oil and gas infrastructure within B.C. Our findings are as follows:*

While we appreciate that many inspections are done annually, the nature of these inspections is not clear to us (are they OGI, volumes quantification, or other?), nor are the results of these inspections visible or open to scrutiny in terms of methodology quality control, etc. Furthermore, the relationship between these inspections, and the provincial inventories, is also not clear. Are the inventories updated on the basis of these measurements? While we do know the OGC is very active, and that its people are working in the best interest of environmental protection, we can't measure our study in relation to these inspections because they are neither visible nor open to evaluation.

For reference, in our campaigns we sampled more than 1,740 pieces of infrastructure in triplicate. In other words, we sampled 5,238 locations. This number of "inspections", collected in under a month, is comparable to the BC OGC annual total. The BC OGC might therefore consider mobile surveying as a supplementary way to collect more data on infrastructure (more passes, more visits, or other) with the same amount of effort. Truck pre-screening would allow the OGC to target its use of OGI and other more time-intensive methods, and to use it for emitting infrastructure only – rather than spending considerable effort to find that no emissions exist. Since the BC OGC has legal access to the well pads and facilities in question, its staff members are also in a favourable position to overcome many of the methodological uncertainties that are communicated within their comments. We would always prefer our surveys to be on-pad if possible because a full pass around the infrastructure provides definitive upwind and downwind data - all in close proximity where concentrations are high. We would be happy to assist the BC OGC where necessary to find an optimal balance between measurement methodologies, and we are presently working with operators on projects similar in theme.

*Overall:*

- *Location of infrastructure: The facility data downloaded from the BC Oil and Gas Commission has NTS or DLS coordinates which are accurate to approximately 400 by 400 area. The discussion paper should provide clarity on whether the NTS or DLS locations were used or if and how the study refined the locations.*
  We obtained shapefiles with locations of both wells and facilities from the online BC OGC Open Data Portal, which was publicly accessible directly before and after this field campaign took place. Both of these shapefiles (wells and facilities) were projected in BC Albers (ESPG 3005) and recorded as point locations. None of the locations in the infrastructure

inventory we compiled from the BC OGC Open Data Portal used NTS or DLS coordinates. Furthermore, we used aerial imagery to verify point locations, the majority of which were located on well pads. And although we could not verify the identification numbers or statuses of the infrastructure during our mobile surveys, we did verify the locations of infrastructure when it was visible from public roads. For additional information please see our response to comment from Anonymous Referee #1 *p9 1.7-8.*

- **Emissions attribution:** *There are numerous situations where multiple permits are issued by the Commission at the same general physical location. The discussion paper does not address how this was handled. When a methane plume is detected the discussion paper should indicate how this is attributed to a source when multiple wells and facilities are attributed to the same geographic location. How was a single release anomaly tied to estimating releases that could be tied to multiple permits at the same physical location?*
  In section *3.2 Emission Sources and Trends* we discuss the potential for inaccurately tagging infrastructure as emitting due to the wide radius (500 m) that had to be used because we were surveying from public roads. In this section of the manuscript we clarify that our analysis includes "probable emitting infrastructure, *plus* possibly emitting co-located infrastructure".

- **Emissions rates may be overstated due to the use of averages:** *In calculating emissions, the STFX/DSF study assumed, even for facilities that had emissions detected just over 50 per cent of the time, that their leak rate was constant and ongoing. The study noted that, especially with venting emissions, the release of methane may not be constant. This assumption has high potential to lead to an overstatement of methane emissions.*
  We only included the persistent emission sources we encountered so that we were providing a conservative estimate of CH4 emission sources in the area. We did not include the episodic emitters in our volume calculations. We combined the fraction of persistent emission sources with our minimum detection limit (g/s) to estimate the total emission volume, which makes it highly likely that this is an underestimation of the total emission volume in the area. Furthermore, we did not include emissions from flowback and liquid unloading, which are likely very large contributors to emissions in an unconventional natural gas development. As described in Allen et al. (2013), these operations have proved to be very large emission sources in these types of developments, but without prior knowledge to when these events were happening we could not include them in our mobile surveys.

*Specific discrepancies within the text are as follows:*

*Page 8 line 22 Well status of:*

- *"Cancelled" means the well permit expired without drilling commencing. So these wells do not physically exist in the field and can not be attributed to the release of methane.*
- *"Well Authorization Granted" (WAG) means that a well has been approved, but drilling has not commenced. Therefore these can not be attributed to methane releases.*

In both our field surveys as well as the independent study by the David Suzuki Foundation (which was submitted to the BC OGC), multiple locations with wells and/or facilities that were classified as Abandoned still had infrastructure standing. So it should be noted that the infrastructure status information was not always correct. Please also see our response to comment from Anonymous Referee #1 *p9 1.7-8* for revised text we have now included in the manuscript.

Although these emission sources might not have been in place at the time of surveying, we are confident that a persistent plume exists at each of those locations. In the manuscript (Section *3.1 Measured Gas Signatures*) we are clear that confidence is high for detection of plumes, but comparatively low for geospatial attribution. Plume detection confidence is high in part because of the excess ratio approach, but particularly because of the persistence requirement in this study where an emission must have been observed > 50% of the times it was surveyed, which was normally on different days. The manuscript also already describes how we benchmarked our rate of false positives using a Control route to validate our level of certainty around detection.

Despite our confidence in detection, the attribution of those plumes to known infrastructure during on-road campaigns can be imperfect. Local wind eddies can serve to complicate back-trajectory analysis. Also, emissions originating farther upwind might cause false tagging of a proximal source. The manuscript does already acknowledge that mis-tagging is possible, and we did provide relative confidence values for detection and attribution in section *3.1 Measured Gas Signatures.*

In response to this comment, we did undertake a new geospatial analysis to search for proximal infrastructure at these Cancelled and WAG locations in question, which numbered only 35 in actual emission inventory calculations. In this analysis we searched for source-types (i.e. possible emission sources in our database) within 3 km.  As we expected, there was almost always other infrastructure nearby. Most of the Cancelled and WAG sites were within 1 km of other infrastructure and all but one were within 1.5 km of other infrastructure. We can, in fact, resolve leaks from those distances, given sufficient source strength, and favourable Pasquill stability. In our analysis we had excluded possible sources > 500 m but in these cases it is reasonable

that another nearby source could have been emitting the plumes we observed repeatedly at those locations.

Sources we did not have in our infrastructure inventory may also explain some of the observed plumes. In the region there is an extensive pipeline network that circulates natural gas between pads and facilities.  Since we did not include pipelines and associated sources in our study, we therefore implicitly assumed that pipeline,  and flow line infrastructural leaks were equal to zero – which is obviously not be the case but was a necessary simplification since we did not have these files of these locations. These 'ghost' sources may also explain plumes in these areas where we detected them repeatedly.

To find the actual source of emissions at these locations, we are happy to work with the OGC. As the OGC knows from having accompanied us on surveys in the field, the technique we used excels at localizing emissions quickly - when used for that purpose, and when site clearances are available. We look forward to working with the OGC to help define the source of these emissions and others that may not be resolved well (or quickly) by OGI. An OGI camera is obviously incapable of resolving ground-dispersed emissions such as pipeline leaks, or low-level plumes coming from infrastructure farther upwind – all of which we can detect. We feel that mobile approaches could enhance the efficacy and efficiency of BC OGC measurement and oversight operations, and we look forward to more conversations in the future on the topic.

*Page 8 line 23*

*It is difficult to understand how the text "for the class defined in the databases as Well Authorization Granted, most of which were somewhere in the stages of development during our visits" could be correct. While some wells with a status of WAG would have commenced drilling between the time the well data was acquired in July 2015 and the study completed Sept. 5, 2015, this number is quite small compared to the total number of wells with a status of WAG. While it is unclear when in July 2015 the researchers obtained well data from the Commission, if we assume the data was obtained on July 1, 2015, there were 1,797 wells with a status of WAG. Between July 1, 2015 and Sept.r 5, 2015, 146 of these wells commenced drilling. As this data is for all of northeast B.C., a subset of these wells are located in the study area. In any event, a maximum of 8 per cent of WAG wells were somewhere in the stages of development during the field visits and the remaining 92 per cent did not physically exist at the time of the study and therefore were incapable of emitting methane.*

*In conclusion, for page 8 line 22 the text should be revised from "25% for Cancelled" should indicate no releases from cancelled and "27% for well authorization granted" should read close to zero for well authorization granted.*

We have changed the following line in the manuscript:

"We calculated an emission frequency of 26% for Abandoned, 25% for Cancelled, 30% for Completed, and 27% for the class defined in the databases as Well Authorization Granted."

Please see our response to comment *43. Page 8* from Anonymous Referee #3 for the text we have added to clarify status type definitions, as well as our explanation for why we included well locations with statuses of Cancelled and WAG.

*Page 9, line 5*

*The text refers to a category of "Undefined". It should be noted the term "Undefined" is not used to describe the well status (Well Authorization Granted, Drilling, Cased, Completed, Active, Cancelled, Suspended, Abandoned). "Undefined" is used to describe the well operational status (Production, Injection, Disposal, and Observation). For example, a cased well would have an operational status of undefined since it was never completed. In addition, undefined is used for the well fluid type (Gas, Oil, Multiple Gas, Multiple Oil, Multiple Oil and Gas or Water) if a well has not flowed in order to define the fluid type. For example, a well that was completed, but did not flow when tested would have an undefined fluid type. An active water disposal well would have a status of ACTIVE WATER DISPOSAL, not UNDEFINED.*

In this section of the text we refer to Figure 6 (Figure 7 in the revised manuscript), which is a plot of the emission frequencies based on operation status (including Production and Undefined wells). We did not include Injection, Disposal, or Observation wells in our emission frequency analysis because our sample size was low. We have revised the text in this section of the manuscript to clarify this and to refer to these descriptions as the operational statuses of the wells.

"A portion of the wells had operational statuses of Production wells, and another portion as Undefined. Only Active Production wells were predictable emitters, with high statistical coherence from route to route (Fig. 7). We did not have a high enough sampling frequency of wells with other operation types (such as Injection, Disposal, and Observation wells) to delineate emission frequencies so we excluded them from the analysis."

*Page 11 Line 11 to 18*

*The development of the MDL or release rate in the study involves significant uncertainty which is not adequately discussed in the text. Further information should be provided on the laboratory experiments used to determine a mean level of dilution of 70 per cent to demonstrate "realistic field conditions" and should include the range of results from those experiments.*

The MDL is established with a standard Gaussian technique similar to that of OTM 33A and others. These methods have been used extensively by industry and academics for nearly half a century. The dilution experiments are extremely straightforward. They consist of exposing the analyser, in a configuration like the field, to different durations of known standard concentration, and to calculate the % dilution. Dilution fraction is a function only of pump rate and cavity size. These analyzers control flow rate extremely closely, and of course cavity size does not change – which means that these offsets are highly repeatable. The process is similar to calibrating a piece of lab equipment – relating peak height to actual concentration under a tightly controlled flow regime. It is a form of calibration that is part of instrument use for an experienced user, and scientific manuscripts will assume that these checks have been done – but these procedures don't generally merit description in the peer review literature.

*Page 11, line 19 to 32*

*NOAA states that the Gaussian dispersion model is recommended as a teaching tool to understand basic concepts and does not recommend its use for dispersion studies. This paper should answer the question as to why this particular model was used when there are a multitude of other dispersion models to choose from.*

*Regardless of the dispersion model used, a sensitivity analysis should be completed for the main inputs used for the analysis in this study. As currently written, it is unclear which meteorological inputs (wind speed, wind direction, temperature, etc.) the researchers used, and whether they were representative of the region. Dispersion modelling can be highly sensitive to input parameters, and as such a further discussion of this uncertainty should be included, especially as the outputs from this modelling are used to determine as the release rate and to estimate a regional emissions inventory.*

*In conclusion, for Page 11 (lines 11 to 32), the technique used to develop the emission factor of 0.59 g/s is questionable.*

The primary purpose of the paper was to determine emission frequencies, not to create a highly accurate volumetric inventory. In crafting this response we moved to using the Gaussian equations directly, since we have existing projects in which they are being used. They provide the same numbers as the NOAA tool, and while the NOAA tool is useful for teaching because of ease of use, that does not make it inaccurate. In our study we have provided a minimal realistic inventory. The fact that it compares very closely to an independent regulator-commissioned study conducted within a comparable timeframe (GreenPath, 2017), provides validation for our work.

The meteorological inputs for the dispersion model were measurements recorded at 1 Hz frequency by the anemometer on our mobile surveying vehicle. We have added text to section *3.3 Minimum Detection Limit* to clarify

that these are the values we used as inputs to the dispersion model.

"The NOAA dispersion model computed the mixing depth using the wind speed, wind direction, and weather data we collected from our anemometer at 1 Hz sampling frequency throughout our surveys."

*Page 12, line 20*

*The term "facility" in the Omara study refers to the sum of wells and equipment at a multi-well site. Facility type as outlined in Figure 8 of this study is not the same as defined in the Omara study. There is no basis for using the emission factor 2.2 g/s in this discussion paper.*

Please see our response to Comment 1 from the review by Brian Crosland. We would also be interested in learning the BC OGC's estimate of facility emissions for the study area.

**Conclusion and Recommendation**

*The fact significant quantities of emissions were attributed to wells that do not exist (i.e. 25 per cent of cancelled wells were reportedly emitting) calls into question the accuracy and validity of the discussion paper. Also, the basis for determining emission factors used in this discussion paper is highly questionable - therefore, this study should not infer that the estimates constitute an emission inventory that could be compared with what is reported under the Greenhouse Gas Emission Reporting Regulation. The Commission would welcome further dialogue to improve this study prior to publication.*

We too would like to work together. New proposed Canadian federal regulations strongly move the industry toward measurement (up to 3x annually per piece), and altogether away from estimation models / emissions factors. This will be a change for everyone. In this new scheme, new sources of data (mobile, satellite) will make oversight easier. These tools are evolving rapidly, and inevitable public availability of such data will force more transparency. It will push not only companies, but also regulators, to step up their game as measurement experts. The industry has been relatively dogmatic in its use of monitoring technology, but must look at the new options, of which many good ones already exist. We would offer that the costs of oversight and compliance could be defrayed significantly by combining methodologies in sensible ways – along the way acknowledging the strengths and limitations of these various methods. As a university laboratory, we are available, willing, and eager to help in this type of research. We thank the OGC for its response, and hope we can work together in the near future.

---

## Author Response (AR2)

We would like to thank the editor for their comment on our response to the reviews, and for their decision to reconsider our manuscript after minor revisions. Below is our response to the editor's request for further explanation of the sources of uncertainty. Changes to the manuscript are shown in colour. We believe the additions, noted below, have resulted in an improved version of the manuscript.

*Dear Authors -*

*The responses to the reviews are, for the most part, complete and comprehensive. However, reviewers 2 and 3 both requested discussion of uncertainties in the analysis. In response, the authors discuss the similarity between their outcome and the outcome of a parallel study. While this is indeed helpful information for validation, it does not address the reviewer concern of uncertainties in the analysis.*

*Please address this concern more specifically in a minor revision, either by providing a quantitative estimate of uncertainty in the approach, or by adding a clear explanation of sources of uncertainty to the analysis and how future work might quantify that uncertainty.*

In order to address the editor's very reasonable request, we have made several changes and improvements:

A. We have aggregated our existing text on the topic of uncertainty. Previously, we had discussed uncertainty at various points in the manuscript, which made it easier to miss these details, and probably served to dilute our message. By aggregating this discussion we can discuss the uncertainties in a more systematic fashion.

B. We added quantitative detail to existing discussions around uncertainty. Where we had alluded to a number, or a level of uncertainty, we added a number.

C. We performed additional Gaussian plume dispersion analysis to provide upper and lower bounds on our minimum detection threshold / volume estimation.

D. In our aggregate section on uncertainty, we also discuss how sources of uncertainty may affect one another.

Our new aggregated section on uncertainty is as follows.

[revised manuscript text omitted]